

# Assessing impacts of dike construction on the flood
# dynamics in the Mekong Delta
**Dung Duc Tran[1, 5], Gerardo van Halsema[1], Petra J.G.J. Hellegers[1], Long Phi**
**Hoang[2], Tho Quang Tran[3], Matti Kummu[4], and Fulco Ludwig[2]**
[1] Water Resources Management group, Wageningen University, P.O. Box 47, 6700 AA
Wageningen, The Netherlands
[2] Water Systems and Global Change group, Wageningen University, P.O. Box 47, 6700 AA
Wageningen, The Netherlands
[3] Southern Institute for Water Resources Planning, 271 An Duong Vuong Street, District 5,
Ho Chi Minh City, Vietnam
[4] Water & Development Research Group, Aalto University, P.O. Box 15200, Aalto,
Finland
[5] Centre of Water Management and Climate Change, Vietnam National University Ho Chi
Minh City, Ho Chi Minh City, Vietnam
Corresponding to: D. D. Tran (dung.ductran@wur.nl)
**Abstract**
There is a large public concern about recent increases in flood risks in the downstream part of
the Vietnamese Mekong Delta. Pronounced recent expansion of high-dike constructions in the
floodplains of the upper delta are associated with observed increases in river water levels in the
downstream province of Can Tho. In this paper the hydraulic impact of upstream dike
construction on the flood hazards downstream the Mekong Delta is assessed through modelling
of dike density scenarios in the flood hydrographs of 2011 and 2013. To do this, the existing
Mekong delta one-dimensional (1D) hydrodynamic model was combined with a quasi-two



dimensional (2D) approach to explore the change in water dynamics downstream under
extensive high-dike developments in An Giang and Long Xuyen Quadrangle. Most studies are
unable to explain where the floodwater goes. To address this, water balances have been
established to trace the propagation and redistribution of flood volumes over the delta, which
provides an extension of current work in this field. Model results indicate that the extensive
construction of high dikes in the upstream floodplains have only limitedly impacts upon the
downstream peak river water levels in Can Tho. Instead, model impacts on peak river water
levels are concentrated and amplified in the upstream reaches of the delta. The water balance
analysis shows the model is able to return fairly stable river water levels at Can Tho by diverting
floodwater volumes away for the Long Xuyen Quadrangle in excess of the retention volume
lost due the dike construction. This reduced inflow into the Quadrangle, and subsequent
diversion of flood volumes to the Plain of Reeds and Cambodian floodplain can, however, not
be fully validated due to a lack of monitoring data. The model's spatial re-distribution of flood
volumes can be induced by the way the model is calibrated. As dike construction results,
according to the model, in a reduction of floodwater volumes reaching the Long Xuyen
Quadrangle, high-dike scenarios for the whole of An Giang province or for the entire Long
Xuyen Quadrangle indicate only limited increases in upstream and downstream peak river water
levels. Future assessments will have to be conducted on the scale of the entire Cambodia-
Vietnamese Mekong Delta, with explicit calibration and validation of the Cambodian
floodplain as a means to trace the re-distribution of flood volumes and peaks across the delta.
**Keywords:** dike, flood dynamics, floodplain, Long Xuyen Quadrangle, Mekong Delta,
hydrodynamic modelling



## 1. Introduction

The Vietnamese Mekong Delta (VMD) is considered the rice bowl of Vietnam, providing about half of the national food volume. Paddy rice is the primary cultivation for about 60% of the people living in the delta. The delta provides the country with 51% of paddy-rice production, 55% of fisheries and fruit production, and 60% of exported aquacultural goods. Food policies stimulating rice production in the delta have made Vietnam one of the world leading rice exporters (Käkönen, 2008). Such a high food production is the result of the high fertility of the delta in combination with abundant (flood) water resources.

Severe floods have negative impacts on the livelihood of local population. A number of severe floods have occurred in the VMD over the last 30 years; notably in 2000, 2001, and 2002 and the latest one in 2011. The 2000 flood is considered the worst disaster of the past 70 years. This flood affected 3.4 million people in 22 provinces, inundated 100,000 ha of paddy fields, and caused 347 casualties of which 80% were children. The economic cost of physical damage alone was estimated at US$ 157 million (Sok et al., 2011). (UNICEF, 2000) estimated the total economic loss to be US$ 257 million. During the 2011 flood, about 89 people died and 40,000 hectares of rice was inundated with an estimated economic loss of US$ 209 million (Cosslett and Cosslett, 2014).

Floods also bring lots of benefits to the delta. According to Tri et al., (2013) and Marchand et al., (2014), about 160 million tons of fluvial sediment are transported annually during flood events which is higher than the estimate of 67 million tons explored by Lu et al., (2014). About 1.86 tons of fish, worth US$ 2.6 billion, was supplied by the flood in 2000. Flood waters also improve soil quality by cleaning rice fields from the (over)use of chemicals and contribute to wetland protection and biodiversity conservation. In general, extreme flood events threaten people and properties whereas small to medium floods bring fertile sedimentation and fish





sources to create an optimal environment for agricultural livelihoods (Käkönen, 2008 and Hung,
2012). The people in the delta thus continuously strive to obtain the maximum benefits from
floods while securing live and properties against extreme flood damages (Wesselink et al.,

4    2015).

Vietnamese frequently change their farming practices to exploit the benefits from floods (Ngan
et al., 2017). Historically, people adapted their livelihoods to floods and its benefits by applying
various flood-based agricultural activities and techniques. For example, floating rice (*lua mua*)
was cultivated that grew along with rising flood water combined with catching natural fish
(Käkönen, 2008). The economic reform policy of *doi moi* in 1986, and the food security priority
to attain self-sufficiency in rice, have had  major influences on the delta's socio-economic
development (Kingdom of the Netherlands, 2011 and Toan, 2011). This has resulted in land-
use dynamics of gradual and progressive intensification of rice cultivation (Sebesvari et al.,
2012). First with the reclamation of the Long Xuyen Quadrangle (LXQ) and Plain of Reeds
(PoR) with semi-dikes (August dikes) and irrigation and drainage canals, enabling cultivation
of two rice crops before a delayed mid-August flood. In 1996, the flood reclamation and
protection program was further intensified with construction and resettlements of residents in
flood protected villages (Danh and Mushtaq, 2011). In this period, the first large-scale flood
control structures were built. During the past decade, high dike compartments have increasingly
been constructed in the LXQ and PoR floodplains, enabling permanent flood protection and
triple rice cultivation.
Today, the floodplains in the VMD are occupied by a large number of compartments (polders)
enclosed by semi-dikes and high-dikes. These changes and intensification of land use in the
floodplains have coincided with the perceived and experienced increases in flood risks
downstream the delta around the city of Can Tho. The water levels observed downstream in
2011 were much higher compared to the situation during the severe flood of 2000. Specifically,





the water level observed in 2011 at the upstream station of Tan Chau was 0.63 m lower than in
2000 (4.27m and 4.90 m), whereas the water levels observed in Can Tho were 0.36 m higher in
2011 than in 2000 (2.15 m and 1.79 m, respectively). This thus suggest there is a clear
correlation between the increase in high dike construction in the floodplains and the increase of
water level and flood risks downstream in and around the city of Can Tho. This paper seeks to
assess and quantify this impact through the hydrodynamic modeling of the high dike
developments within the flood season hydrographs of 2011 and 2013.
Several studies have concluded that flood risks in the delta have increased over the last decades,
and associate them with a number of reasons: climate change, sea level rise, hydropower
development, land subsidence, and local rainfall (Wassmann et al., 2004; Hoa et al., 2007; Lauri
et al., 2012; Tri et al., 2012; Fujihara et al., 2015). In the complex hydrodynamic context of the
delta, flood risks are assessed in various forms by a variety of authors: flood extent and duration,
flood depth or floodwater level, or river water levels. Each of which is associated with and
influenced by specific hydrodynamic characteristics and parameters that need to be accounted
for. Fujihara et al. (2015) conducted a study to identify the possible impacts of the runoff
upstream, sea level rise, and land subsidence on the floodwater level rise, based upon the
analysis of observed daily minimum, maximum, and mean water levels of 24 monitoring
stations from 1987 to 2006. The authors concluded that the flood depths significantly rise in the
tide-dominated areas and the inundation is strongly impacted by land subsidence (6.05 mm
year$^{-1}$~80%) and sea level rise (1.42 mm year$^{-1}$~20%). Using the VMod hydrological model
with statistical downscaled methods, Lauri et al., (2012) assessed the impacts of projected
climate change and reservoir operation on the future changes in the Mekong River hydrology.
They identified that the changes in discharge at Kratie due to planned hydropower operation
are higher than climate change impacts. Some other studies used different 1D hydrodynamic
models such as ISIS and Mike 11 to evaluate the impacts of climate change and sea level rises



on flood propagation, flood inundation and sediment transport in the VMD (Apel et al., 2012;
Hung, 2012b; Quang et al., 2012; Tri et al., 2012, 2013; Manh et al., 2014).
Some studies identified and assessed changes in water levels in the VMD due to the effects of
infrastructure scenarios. Specifically, Hoa et al. (2008) used the HydroGIS hydrodynamic
model, based on the analysis of the measured data from 1961-2000, to evaluate hydraulic effects
of infrastructural changes (1996-2004) on floods under flood prevention activities. The authors
concluded that the infrastructure changes as dredging canals, raising some important
embankments, and upgrading roads can mitigate the flood's extent but increase the flood depth
by 0.2-0.3 m in some regions near and between the embankment systems. Also using the
hydrodynamic model of Mike 11, Duong et al. (2014) simulated the water level changes
impacted by the full-dike constructions of the floods of 2000 and 2011. They identified that the
water levels increase by 13 cm at Chau Doc and 5 cm at Can Tho in Tien River under the
scenario of the flood 2000 on the network 2011 in comparison with the flood 2000, and -8 cm
at Can Tho for the 2011 flood on the 2000 network, but they were unable to explain how and
where the flood water distributes. Dung et al. (2011) noted that the hydrodynamic model
contained deficiencies in the representation of the dyke system in Vietnam.
Large-scale constructions of dike compartments might reduce the flood retention capacity
because these compartments prevent floodwater entering farmlands. Marchand et al. (2014)
proposed that the higher recorded river water level downstream during the flood of 2011
compared to the 2000 could be attributable to the construction of higher dikes. Fujihara et al.
(2015) indicated the need for more research on the impacts of high-dike constructions on water
regimes. This study addresses this knowledge gap using a modelling approach to test the
hypothesis that large-scale high-dike constructions would reduce the flood retention capacity
of the floodplains and cause increased water levels (and associated flood risks) along rivers
downstream.





This paper aims to identify the impact of dike constructions on flood dynamics, focusing on
changes in river water levels and spatial distribution of floods in the VMD floodplain. To do
this, first we further developed and calibrated a hydrodynamic model for the entire delta to
simulate floods under different dike construction scenarios (Sect. 3). Second, floodwater
balances were calculated from the simulation results to identify and quantify changes in flood
dynamics. On the basis of the modelling results we analyse how dike constructions lead to
changes in flood pattern and river water levels across the floodplain (Sect. 4). We finally discuss
the main findings and uncertainties (Sect. 5) and draw conclusions (Sect.6).
**2.  Study area**
The Mekong Delta (MD) consists of the VMD and the floodplain in Cambodia, covering a
region of 5 million hectares. It is defined from Kratie in Cambodia and the main river flows
through the Cambodia floodplain (see Figure 1). At Chaktomuk, the river is separated into two
watercourses: the Tonle Sap diverts (wet season) & abstracts (dry season) water to and from
the Tonle Sap Lake and the other enters Vietnam via two branches, the Mekong River (Tien
River in Vietnam) and the Bassac (Hau River in Vietnam) (Manh et al., 2014). The Tonle Sap
Lake plays an important role as a natural reservoir mitigating flood flows and ensuring dry
season flows downstream for the Tien and Hau Rivers to distribute the large amount of water
to canal systems in the VMD (Kummu et al., 2014).
Located in the North Pacific monsoon climate (Tamura et al., 2010; Manh et al., 2014), the
delta is strongly impacted by flooding upstream and the tides from the Gulf of Thailand and the
South China Sea. Floods occur during the wet season from July/August to
November/December. When the annual average discharge at Kratie exceeds 13,600 $m^3s^{-1}$, the
flood season in the MD starts (Manh et al., 2014). At Tan Chau, the Tien River carries about
80% of the water (equivalent to 20,500-25,500 $m^3s^{-1}$) during the flood event whereas 20%




(equivalent to 6,500-7,660 m$^3$s$^{-1}$ at Chau Doc) is transported by the Hau River (Tri, 2012). But
after Vam Nao, the water volume of the two rivers is relatively balanced owing to the
interconnected tributaries. Due to its flat and low-lying topography of average elevation of 0.8
m a.m.s.l and the impact of tidal regimes (Hung, 2012b), a flood yearly inundates about 1.2-1.9
million hectares of the delta (Hoa et al., 2008; Mekong Delta Plan, 2013b). Severe floods
seasonally cause a water depth of up to 3 m and affect lives of more than 2 million people. Due
to the flood and tidal impacts, the delta has a complicated flow regime.
The LXQ and PoR play important roles in mitigating flood peaks thanks to their large capacity
to retain floodwater during the flood season. This floodwater originates from the two main
rivers and the over-land-flow from the Cambodia border. Seasonal floods provide benefits to
the delta in the form of sediment and nutrient deposition, land flushing (e.g. sulphites, acid and
agro-chemicals) and rich fish sources (Howie, 2011; Danh and Mushtaq, 2011; Hung,
2012a&b). However, severe and extreme floods also impose flood risks with potential damage
to the economy and loss of life. The construction of embankments and dikes equipped with
sluice gates started on a large scale in the 1990s, with the aim to protect residential areas and
the intensification of rice production (Käkönen, 2008). The government's food security policy
actively supported development of double and triple rice cultivation in the floodplain. To date,
most agricultural areas in the floodplain are constructed with August or high dike
compartments.
We focus on the LXQ in this study as it has been the portion of the VMD floodplain that has
undergone the most extensive development of high dike compartments over the last decade. It
is therewith also one of the six central economic zones of the VMD with the highest productivity
in rice  (Quang et al., 2012). The 0.49 million hectares floodplain is located in the Northern part
of the delta at the right bank of the Hau River. The LXQ consists of large parts of An Giang,
Kien Giang provinces and a small part of Can Tho province (see also Figure 1). There are a





large number of dike compartments in between the dense river and canal networks, including
roads and dikes. Currently most agricultural areas in the rice dominated LXQ are protected by
two types of dikes. There are semi-dikes allowing floods to overflow into the field after harvest
of the second crop in mid-August (so called August dikes), and high-dikes preventing floods
for the whole year, enabling cultivation of a third rice crop (Howie, 2011). Statistical data of
the Department of Agricultural and Rural Development show that the high-dike areas increased
from 2,591 hectares to 87,909 hectares in An Giang province between 1998 and 2009 (Kien,
2013). In Kien Giang province, semi-dikes protect most of the agricultural areas and there are
hardly any dikes in Can Tho province. Large construction of high dikes reduces the flood
retention capacity of the floodplains and can potentially cause higher flood risks downstream.

11                                    **[Figure 1]**

**3.  Methodology**
**3.1.The model setup and data preparation**
The one-dimensional hydrodynamic model based on Mike 11 software developed by the Danish
Hydraulic Institute (DHI), was used to represent the river network and its floodplains for the
Mekong Delta. The model is a numerical model using equations of Saint Venant and Continuity
Equation for solving complex flow and mass transport problems (Patro et al., 2009; Dung et al.,
2011; Manh et al., 2014). Detailed information about the equations and computational
components is available from DHI, (2011).
The river network and related physical data of the MD was derived from the Southern Institute
for Water Resources Research (SIWRR). We applied the Hydro Dynamic (HD) module to
establish the simulations of flow dynamics and inundations. Basically, it includes 4 main
components: (i) river network, (ii) boundary condition, (iii) cross-section, and (iv) parameter





set. Although the rainfall only accounts for a low percentage of surface water inflow, it was
also considered in the model using the Rainfall Runoff (RR) module.
The river network of 2011 was setup into the model, based upon available data, to cover the
domain of 5 million hectares. It consisted of 4,084 branches and 21,235 of computational nodes
to describe the river system from Kratie and Tonle Sap Great Lake in Cambodia to the river
mouths in Vietnam. In the network, sluice gates (14), weirs (2,246), and control structures
(2,657) were set up to represent the infrastructure systems. The sluice gates were established at
important points where their functions can control water flow for a large area. The control
structures were considered as reservoirs that prevent water overflow at a specific sill level while
the weirs regulate water flows in and out of the compartments.
The required boundary conditions for the model include discharges and water levels observed
in 2011 and 2013. All data, at daily intervals, was provided by the National Centre for Hydro-
Meteorological Forecasting (NCHMF) and the SIWRR. The discharges of 6 stations were set
up for upstream boundary conditions while the water levels of 9 tide gauges near the coast were
used for downstream boundaries. Upstream, the discharge at Kratie is the most important
boundary input for the setting of the main flood hydrograph for the VMD.
13,000 cross-sections were embedded into the model to describe the topography of rivers and
branches. Those cross-section data were collected from various sources. The data for the
mainstreams have a higher accuracy because their measurements were regularly updated from
national projects. Cross-sections of most branches were measured in the bathymetric data, so
the accuracy is lower. However, these cross-sections were tested over different projects of the
SIWRR.
The parameter set includes river roughness, wind effect, and various components that could be
read further from DHI, (2011b) were put into the model to describe the physics of the MD.



Among them, the river roughness coefficient is the most important and sensitive parameter in
model calibration process for water level simulations. The initial river roughness were
estimated based upon the range of roughness values corresponding to types of rivers and canals
provided by various publications (Chow, 1959; Fabio et al., 2010; Dung et al., 2011). These
roughness numbers are represented as manning coefficients in the model. We first set global
manning coefficients of 0.02, 0.025, and 0.033 for the initial runs to identify the changes in
water levels and discharge in the main rivers. Second, the model was calibrated by changing
these numbers in branches near the coastal area. After the model fitness is accepted for stations
near the coast, the range of manning coefficients (0.024 - 0.017) for the Tien and the Hau Rivers
was defined. Rivers in the Cambodia part were set with a range of (0.1-0.05) whereas a range
of (0.03-0.025) was selected for rivers and canals in the floodplains of the VMD. These
parameters were optimized during the calibration.
Daily rainfall data from 37 meteorological stations (28 in Vietnam and 9 in Cambodia) were
used to run the model. Thiessen polygon technique was used to describe the contribution of
surface water on rivers and canals. In the model, the Mekong Delta was divided into 120 sub-
regions and each sub-region was distributed rainfall water based upon each responsible rainfall
gauge. The rainfall discharge had to be calibrated from NAM rainfall-runoff component
coupled with Mike 11 before it could be used for the hydraulic model simulations.
**3.2. Calibration and validation**
The flood model was calibrated and validated to ensure reliable performance. We calibrated the
model with the extreme flood year 2011 and a validation was carried out using data from the
2013 flood season. These two years were selected because network, land use and dike locations
are relatively similar in both years. An analysis of the Nash-Sutcliffe efficiency and the
correlation coefficient was carried out to check the goodness-of-fit measures of the model for





the calibration and validation periods. Nash-Sutcliffe efficiency (NSE), one of the most
commonly used in hydrology, was selected as an efficiency criterion to measure how much of
the variability in observation is explained by the simulation. The simulation is perfect if the
NSE value reaches 1 (Ritter and Muñoz-Carpena, 2013). The correlation coefficient ($R^2$)
expresses the linear dependence between observed and simulated values.
In the calibration and validation periods, we used the hourly discharge and water level time
series observed from 15 gauging stations, including 11 stations along the Tien and Hau Rivers,
and 4 stations in the floodplain (Figure 1). We selected these stations because (i) the objective
of the study was to explore the water level dynamics in the main streams and the LXQ and (ii)
the availability of observed data.
In addition to calibration and validation for 2011 and 2013 data, the model performance was
also assessed for the flood hydrograph of the year 2000. Flow data of 2000, including discharge
at Kratie and the water levels from 9 tide gauges, were used to force the model assuming a 2011
river network and land use system. Model output were used to evaluate the flood dynamics in
terms of maximum river flows in the Hau River.

### 3.3.Modelling for the floodplains

Floodplains in Cambodia and Vietnam deltas were modelled using two different approaches.
The Cambodia floodplain without channels and dikes is simulated by the wide cross-sections
using the 1D method. The quasi-2D approach was applied to formulate the hydrodynamic
interactions between floodplains, and rivers under dike construction scenarios in the LXQ.
Although the Plain of Reeds was not itself the focus of this research, it was also included in the
model with the constructed dike in 2011 because there are important hydraulic interactions
between the Tien and the Hau Rivers through the Vam Nao River and their tributaries. Each
compartment in the floodplains was considered as a flood cell modelled by cross sections





extracted from the digital elevation model (DEM) with 90mx90m resolution, which were
attached into fictitious river branches. These fictitious river branches were linked to real
channels by the control structures. Dikes and overflows were represented by weirs whereas the
dike's height was adjusted by changing the sill level of control structures. The detailed
methodology of the quasi-2D was described by Dung et al., (2011).
**3.4. Dikes construction scenarios**
**[Figure 2]**
Different dike construction and land use scenarios were developed to explore the impacts of
dikes on flood dynamics (Figure 2). The first scenario (S1) is used as a baseline to explore the
flood dynamics without the impact of high-dikes. Therefore, all the high dikes are removed
from the model. Without high-dike compartments, the water discharges are freely distributed
into the LXQ and throughout the canals along the Hau River. The second scenario (S2)
represents the land use and dike conditions in 2011. Here, the high-dikes account for more than
50% of the total agricultural area in An Giang province, with the remaining areas protected by
semi-dikes. Some districts in the province constructed high-dikes, but there are also some
districts were semi-dikes dominate. For example, Kien Giang province had semi-dike only in
2011. The third scenario (S3) represents a system with high-dike protection for the whole An
Giang province. The fourth scenario (S4) represents a system with high dikes for the entire
LXQ.
**3.5. Water balance calculation**
In order to ascertain the reasons why and where the water movement in the floodplain causes
water changes in the downstream part, we implemented water balance calculations for each
scenario. In the 1D hydrological model used for a complex hydraulic situation as the Mekong





Delta, all components in the water balance formulation are estimated. Therefore, based upon
the characteristic of the Mike 11 model, the water balance formulation is as follows:

$$\sum_{i=1}^{n} Q_{in}(t_i) - \sum_{i=1}^{n} Q_{out}(t_i) = (V - V_o)dt_i$$

Where $\sum_{i=1}^{n} Q_{in}(t_i)$ is total inflows, and $\sum_{i=1}^{n} Q_{out}(t_i)$ is total outflows to the LXQ, in cubic
meter per second ($m^3s^{-1}$), corresponding to the starting time $t_1$ (July) to ending time $t_n$
(December) of the flood simulations. V is the controlled volume, and $V_o$ is the initial volume,
in cubic meters ($m^3$).
From the output of the hydraulic model, we extracted discharge time series data from canals
along the closed boundaries of the LXQ to calculate volumes over a period from July to
December. The inflows include the water fluxes along the Vinh Te canal and along the Hau
River, while the outflows were taken from the Cai San canal and the canal along the Gulf of
Thailand. Water balance was also computed for the Hau River. The water fluxes at Chau Doc
and delivered from the Tien River are input flows, while the output flows consist of the water
discharges along the Hau River to the LXQ, through Cai San canal, and the point on the Hau
River after the Cai San canal. Rainfall volumes were calculated from the separated rainfall
simulation files.
**4. Results**
**4.1. Calibration and validation results**
The calibration and validation results are presented in Table 1. Additionally, the Q-Q plots are
presented for representative stations (Figure 3). Generally, the NSE and $R^2$ values computed
for selected stations present a very good performance of the model, respectively in ranges of
(0.79-0.97) and (0.89-0.98). In the calibration period 2011, there is a better performance




compared with the validation 2013. But this is to be expected, as additional changes in
infrastructure and dike construction that may have occurred between 2011 (calibration) and
2013 (validation) are not incorporated in the model. For example, the NSE of the water level
found in Chau Doc in 2013 is just 0.79 compared to 0.92 in 2011. Similar performance was also
identified for My Thuan. Comparisons of the discharges at My Thuan also show lower NSE
values in both 2011 and 2013, but these values are still larger than 0.8. The stations within the
floodplain also show good agreements of water levels in a range of (0.85-0.96); unfortunately,
the observed discharge data are not available for those stations.

9                   **[Table 1]**

10                  **[Figure 3]**

The model performance is relatively good based upon the small difference of the peak
simulation and observations in 2011 and 2013 (Figure 4). However, the simulated peak values
are in most cases lower than the observed data. For 2000 the model bias is higher compared to
2011 and 2013. Model simulations results show a slightly lower peak river water level at Can
Tho in 2000 (2.02 m) compared to 2011 (2.10 m). Observed data show a much higher difference
in peak river water levels for Can Tho. In 2000, the highest water level observed was only 1.79
m compared to 2.15 m in 2011.
The question thus rises whether these changes in river water levels at Can Tho can be primarily
attributed to changes in the floodplains and canal networks that occurred between 2000 and
20011, or to the effect of observed higher tidal water levels in the estuaries of the Tien and Hau
rivers. The observed tidal water levels at My Thanh and Ben Trai in the estuaries (Table 2) are
markedly higher in 2011 than for the peak flood year of 2000, with potential backwater curve
effects on observed and modelled river water levels at Can Tho for 2011. However, as the model
results for 2000 (using the network and dikes of 2011 with the observed river and tidal data of



2000) compared to those of 2011 (network 2011 and water levels 2011) only show an increase
of 0.08 m at Can Tho for 2011 (Table 3), the tidal backwater effect seems limited. It is
significant smaller than the observed difference between 2011 and 2000, which amounts to 0.36
m for Can Tho. This analysis suggest that the effect of tidal influence is approximately 0.08 m,
whilst the effect of occurred changes in the network and floodplains amounts to 0.28 m on the
river water levels at Can Tho. Given that the total flood volume of 2011 was 30 per cent smaller
than that of 2000 ($283x10^9$ m$^3$ : $402x10^9$ m$^3$) the effect of changes in the network and floodplains
is relatively large.
**[Table 2]**
**[Table 3]**
**[Figure 4]**
**4.2. Flood dynamics under the impact of 2011 dike and high-dike construction in An**
**Giang province and Long Xuyen Quadrangle**
Model results indicate that if all high dikes would be removed (S1 scenario), peak river water
levels would be much lower especially in the upper part of the Mekong Delta (Figure 5).
Compared with the 2011 situation (S2) the peak river levels would reduce by 66 cm at Chau
Doc and 31 cm at Vam Nao if all high dikes would be removed. At Can Tho, however, the
difference in peak river levels result to be relatively small; removing all the high dikes reduces
peak water levels by about 4 cm. Inside the LXQ high dike removal reduces peak water levels
more upstream (90, 40. and 50 cm at Xuan To, Tri Ton and Tan Hiep respectively), compared
to downstream points (2 cm at Phung Hiep) (Figure 6). The water levels reduce in the Vinh Te
canal under a no high dike scenario (17.2-84.6 cm from upstream to downstream), but increase
in the Cai San canal (4.3-45.8 cm) and the canal along the Gulf of Thailand (a fluctuation of 1-
34.1 cm along the canal) (Table 4).





The increases in river water levels from expanding of high dikes for An Giang and Long Xuyen
Quadrangle have the same pattern with those in the comparison between the S2 and S1 (no-
dike scenario). The model presents very slight increases (2-3 cm upstream and 1 cm
downstream) in river water levels from expanding of high dikes for An Giang and Long Xuyen
Quadrangle (S3 and S4) compared the 2011 dike (S2) (Figure 5 and Figure 6). Overall, the
major differences are only found between the S1 and the high dike scenarios.

7                                    **[Figure 5]**

8                                    **[Figure 6]**

9                                    **[Table 4]**

**4.3.Floods of 2000 and 2013**
To assess the impact of different floods on the peak river water levels, the scenarios S1-S4 were
run with flood hydrographs of 2000 and 2013 in comparison to the base runs of 2011. These
simulations result in similar upstream concentrated increases of water levels for all three-flood
hydrographs (Figure 7). The largest increase in river water levels are the result of the high dike
scenarios (S2-S4) that return similar absolute increases in relation to the no-dike scenario S1
under all three hydrographs. Thus, suggesting that the peak river water levels along the Hau
River are relatively independent of the flood size and regime, as the flood volumes for the runs
differ quite starkly from $402 \times 10^9$ m$^3$ (2000), $283 \times 10^9$ m$^3$ (2011) and $236 \times 10^9$ m$^3$ (2013).

19                                    **[Figure 7]**

**4.4.Hinge-effect.**
The results of the simulation runs conducted for the four scenarios and the three flood
hydrographs indicate a hinge response operating on the water levels of the Hau River. The
increases in water level are more pronounced (and concentrated) in the upstream reaches,





whereas the water levels at downstream Can Tho are fairly constant across scenarios and across
flood hydrographs (Figure 8). For the scenarios S1-S2 across the hydrographs 2011, 2000 and
2013, this results in a coefficient of variation CV of water levels at upstream Chau Doc of CV
= 0.47 diminishing to a CV = 0.07 at downstream Can Tho (Table 4). This 'hinge-effect' may
be influenced by: i) the use of tidal water level data at the river estuary as boundary condition
into the model; ii) the coast-to-upstream direction of model calibration.

7                                        **[Figure 8]**

The tidal water level data at the estuaries of the Mekong are inevitably influenced by the (peak)
river discharges of the year in question – with peak river flows significant increasing estuary
water level at high and low tide. The model's boundary conditions are thus not only set by tidal
movements, but also influenced by the river discharge at the estuaries for the used years.
Simulation runs are thus forced towards estuary (and river outflow) water levels as recorded for
the year, providing a fixation of the hinge at the estuary.
The model's calibration sequence, where the coastal network's roughness coefficients are set
first to meet the recorded water levels, before the upstream reaches are set, further strengthens
this 'hinge-effect''. Potential errors and deviations of water level are propagated towards the
upstream reaches and outer edges of the model.
On the other hand, the dissipation effect of a large floodplain and network as represented by the
Mekong Delta is expected to yield relatively smaller amplitudes in downstream water levels, as
changes are dissipated across the floodplain and network. However, any subsequent reductions
in the floodplains and its dissipation capacity are then expected to be noticeable on an increased
amplitude of downstream water levels.




**4.5.Water balance**
In order to further ascertain the model's simulation of the hydrodynamic characteristics of the
Mekong Delta, an analysis of the water balance of flood volumes was carried out for the Long
Xuyen Quadrangle (LXQ) floodplain .Compared to the situation without high dikes (S1), the
high-dike scenarios (S2 - S4) result in the reduction of floodwater flowing into the LXQ (Figure
9). The floodwater volume decrease from $18.4 \times 10^9$ m$^3$ to $12.7\text{-}11.8 \times 10^9$ m$^3$ along the Vinh Te
canal bordering Cambodia, and from $31.7 \times 10^9$ m$^3$ to $12.5\text{-}11.3 \times 10^9$ m$^3$ along the Hau River.
With less water coming into the LXQ floodplains the water drained into the Gulf of Thailand
and into the Cai San canal is reduced accordingly in the high dike scenarios (from $33.3 \times 10^9$ m$^3$
to $22.1 - 21.3 \times 10^9$ m$^3$, and from $16.6 \times 10^9$ m$^3$ to $2.4 - 2.6 \times 10^9$ m$^3$, respectively). The total
reduction in flood volume entering the North-West corner of the Mekong Delta amounts for the
simulation runs S2-S4 (2011 hydrograph) to a decrease of $15 \times 10^9$ m$^3$ (Table 5), which amounts
to a reduction of flood volume higher than the estimated flood retention capacity of $13 \times 10^9$ m$^3$
for the entire LXQ (estimated as a flood depth of 3 m over the entire 0.49 million hectares
floodplain). This thus explains how the model simulation runs S2-S4 for high dike construction
return only a minimal increase in river water levels, despite a significant reduction of flood
retention capacity in the LXQ. In the model simulations, floodwater is diverted away from the
floodplain.
The floodwater volumes reaching the LXQ decrease under the high-dike constructions (S2 -
S4). First, the overflow from the floodplain of Cambodia to the Vinh Te canal decreases from
$16.8 \times 10^9$ m$^3$ in the situation without high-dike (S1) to $13.5\text{-}12.6 \times 10^9$ m$^3$ under the high-dike
constructions. Second, the floodwater from upstream the Hau River reduces from $84.1 \times 10^9$ m$^3$
to $81.8\text{-}81.6 \times 10^9$ m$^3$. Finally, the amount of floodwater from the Tien River flowing into the
Hau River decreases from $138 \times 10^9$ m$^3$ to $128.9\text{-}129.3 \times 10^9$ m$^3$. Combined, these diversion of
floodwater flows amount to a reduction of flood volume of $15 \times 10^9$ m$^3$ (Table 5).




The high dike scenarios (S2-S4) result in changes in flow direction of the modelled flood flows
and volumes, resulting in only a slight increase of the flood volume at the downstream (estuary)
reach of the Hau River. In the Vinh Te canal, the flood flow drains an amount of $1.6 \times 10^9$ m$^3$
towards the Gulf of Thailand for the no-dike scenario (S1-2011). For the high dike scenarios
(S2-S4, 2011) it reverses direction diverting $1.3$-$1.4 \times 10^9$ m$^3$ towards the Hau River. In the Cai
San canal, the flood flows into the Hau River with an amount of $4.26 \times 10^9$ m$^3$ but under the
impact of high-dikes changes direction towards the Gulf of Thailand with an amount of $2.74$-
$2.77 \times 10^9$ m$^3$. At the downstream reach of the Hau River the model only returns a slight increase
of flood volume ($0.5$-$1.8 \times 10^9$ m$^3$ ; $< 1\%$) for the high dike scenarios (S2-S4) when compared
to the without dike (S1) scenario. Thus, re-enforcing the "hinge-effect", whereby the volume
of floodwater and water levels at the downstream reach of the Hau River are fairly stable under
all modelling scenarios.  As the water balance analysis shows, this is achieved by a diversion
(re-routing) of flood volumes away from the LXQ, so that the reduction of flood retention
capacity due to increase of high dikes has little impact on downstream water levels and flows.
The reduction in flood retention capacity in the LXQ of $7$-$13 \times 10^9$ m$^3$ (Table 5) is thus
effectively (over)compensated in the model runs by a reduction of $24$-$26 \times 10^9$ m$^3$ of flood
volume entering the LXQ floodplains (Figure 9). This diversion of flood volumes to primarily
the Plain of Reed ($+ 9 \times 10^9$ m$^3$) and Cambodian floodplain ($+6.7 \times 10^9$ m$^3$) by the model can, at
present, not be further ascertained against stage and discharge data due to limitations in the
reach and density of the monitoring network of the Mekong floodplains.

21                              **[Figure 9]**

22                              **[Table 5]**


**5. Discussion**
Recent studies into the flood dynamics of the Mekong Delta have raised concerns over the
increasing flood in the downstream reaches of the Vietnamese Delta as being influenced by the
recent increases in high dike developments in its upper reaches (Hoa et al., 2007; Duong et al.,
2014; Marchand et al., 2014; Fujihara et al., 2015). Using a 1D hydrodynamic model combined
with a quasi-2D approach (following Dung et al., 2011), we aimed to quantify the impacts of
extensive high dike construction scenarios on the changes in flood water levels and risks in the
Vietnamese Delta.
Our modelling assessment shows a clear and distinctive impact on the flood water levels of the
Hau River between the high dike (S2-S4) and no dike (S1) scenarios. For all modelled flood
hydrographs (2000, 2011 and 2013) the high dike scenarios return a marked increase of the
peak river water levels in the upstream reaches of the Hau River (+ 68 cm at Chau Doc), while
minimal increases occur in its downstream reaches (+ 5 cm at Can Tho). A similar trend and
effect is operating on the water levels in the canal network of the Long Xuyen Quadrangle
(LXQ) and western flood plains. The model outputs for high dike (S2-S4) to no-dike (S1)
scenarios show a high impact (+ 100 cm) on water levels along the upstream edge of LXQ (i.e.
at Xuan To), a marked decrease (- 45 cm) within the dike protected floodplain, and a limited to
no effect in the floodplain downstream of LXQ and Can Tho (i.e. at Phung Hiep).
These results are consistent with previous studies that made use of 1D hydrodynamic models
with quasi-2D approaches, and that reported concentrated water level increases (+60 to +100
cm) in the upper reaches (Hoa et al., 2007; Duong et al., 2014; Fujihara et al., 2015), and limited
increase (+ 4-5 cm) in the downstream reaches (Duong et al, 2014) when assessing the recent
extension of high dikes in the LXQ. Thereby, these alerts to an aggravated increase of water





levels and flow velocities in the upper reaches of the delta that are associated with increased
bank erosion and heightened risks for catastrophic dike failures (Hoa et al., 2007).
Our results for the scenarios runs further indicate that the impact of additional expansion of the
high dikes and polders in the LXQ on the peak water levels is limited - simulated with the
scenario S3 (fully diked An Giang) and S4 (fully diked LXQ) -- and only a fraction of the
reported impact between no-dikes (S1) and the 2011 dike situation (S2). This suggests that
additional expansion of high dikes will only have a limited additional impact on river water
levels, and that the highest impact has already occurred with the 2011 extent of dike
construction.
Similarly, the impact of the Mekong flood hydrograph and volume, as simulated with the flood
hydrographs of 2011, 2013 and 2000, is fairly limited on the water levels of the Hau river and
canal network. Although substantially differing in total flood volumes ($402 \times 10^9$ m$^3$ (2000),
$283 \times 10^9$ m$^3$ (2011) and $236 \times 10^9$ m$^3$ (2013)), the impact on the Hau River peak water levels is
very limited in our simulation runs, with the largest amplitude in the upstream reaches at Chau
Doc, but at a fraction of the impact of scenario runs S1-S2. This suggests that the flood volume
and hydrograph of the Mekong has a very limited impact on the peak water levels of the Hau
River.
In both cases – the further extension of high dikes (S2-S4), and the differing flood hydrographs
(2011, 2013, and 2000) – the sensitivity of peak water levels in the downstream reaches of the
Hau (i.e. Can Tho) is very low. A hinge effect is clearly discernible that concentrates the limited
impacts on water levels upstream. However, the larger part of impacts one would associate with
a doubling of high dike protected polders (S2-S4) and Mekong flood volume (2011 – 2000) are
clearly absorbed elsewhere in lower Mekong Delta in our model runs.




Our model runs have a good performance based upon the calibration (S2 – 2011) and validation
(S2-2013) runs, in which the state of high dikes in 2011 (S2) is compared against the recorded
water levels of gauging stations for the respective hydrographs of 2011 and 2013. Consistent
with previous studies, this suggests the model set-up and calibration is performing well in re-
producing recorded water levels. For the model runs S1, S3 and S4 for all three hydrographs,
however, no validation and calibration options were available (as all three represent scenario
settings for extent of LXQ poldering). Our simulation runs in the 2000 flood hydrograph did
not return a neat fit to the recorded water levels in 2000 of the Hau River (Figure 8). Upstream,
scenarios returned lower than recorded values and at downstream Can Tho slightly higher.
Partially, this may be attributed to changes occurred in river and canal network between 2000
and 2011 (e.g. dredging and trimming) that may have altered the hydraulic conducts of the Hau
river, and that have not been captured in our scenarios. The biggest known change in this period,
however, namely the expansion of high dikes (<10,000 ha in 2000 to >140,000 ha in 2011 in
An Giang alone), have been captured in our no-dike scenario (S1). The recorded raising of the
water level at Can Tho (from 1.79 m in 2000 to 2.15 m in 2011/13) over this same period can
also be partially attributed to the siltation of the Hau (reported as Bassac) estuary as projected
by Hoa et al. (2007). According to Hoa et al. (2007) the progressive siltation of the Hau/Bassac
estuary would lead to an increased back water effect as the discharge capacity of the river would
progressively decrease with siltation, sea level rise and storm surges, potentially raising water
levels at Can Tho with up to 100 cm.
In the outset of our study, we expected the expansion of high dikes to lead to higher discharges
in the Hau River, resulting in a more pronounced backwater curve and higher water levels at
Can Tho, as those reported during the peak of the floods in 2011. However, we do not find this
corroborated in our modelling results, where the water level at Can Tho is surprisingly stable,
and all changes in water levels are hinged upstream. The marked increase of flood volume, and





associated river discharge, of the 2000 hydrograph in comparison to that of the 2011/13
hydrographs, is not resulting in a marked change in modelled water level at Can Tho once would
expect with a back-water effect operating on the Hau estuary. The relative stability of the water
level at Can Tho, can only be explained by a relative stability in discharge operating upon the
lower reaches of the Hau.
Our analysis of the water balance of flood volumes – conducted for the hydrograph of 2011 for
scenarios S1-S4 – show how in the model simulations the water is re-distributed over the delta.
Where the simulated water levels of the Hau River and VMD-network operate within a narrow
range, the simulated floodwater volumes diverted throughout the delta operate within a wider
range – both in terms of volume and flow direction. In essence, this is directed by the mass-
balance equation underlying the hydrodynamic model: as the range of water level fluctuation
sis limited by the 'hinge-effect', changes in hydrographs and flood retention capacity of
floodplains are accommodated in the changes in flood volumes and flow directions. In order to
be able to return fairly stable water levels in the downstream reaches of the Hau River (Can
Tho and estuary), the reductions in flood retention capacity in the LXQ (e.g. scenarios S2-S4),
are compensated by a reduction of floodwater volume entering the system under analysis and
floodplains of LXQ ($\Delta$ Storage LXQ -7 to $- 13\text{x}10^9$ m$^3$; $\Delta$Q entering the western delta –
$15.8\text{x}10^9$ m$^3$; and $\Delta$ Q entering the LXQ plain $-26\text{x}10^9$ m$^3$). Thus, this enables the model to
return a fairly stable water level and floodwater volume ($195\text{x}10^9$ m$^3$ ± 1%) in the downstream
reach of the River. The scenario runs indicate that the impacts of flood retention loss in LXQ
due to high dike constructions are mainly concentrated in the upstream and eastern reaches of
the delta, whilst having limited impacts on the downstream reaches of the Hau River and Can
Tho. The increases in flood volumes and risks are in the simulation runs primarily re-directed
towards the Tien River and the Plains of Reeds and the Cambodian floodplain. Whereas this
may be an outcome of the present model configuration there are at present no means of





verification available (in the form of water level and discharge data) to validate and calibrate
such changes in flood volumes and directions towards the Plain of Reeds and Cambodian
floodplain.
The hydrodynamic model approach applied could also have influenced the accuracy of flood
simulation and water balance equations. At small scale, two-dimensional and three-dimensional
hydrodynamic models (2D and 3D) are the most suitable options to simulate food dynamics of
a complex floodplain. However, the 2D and 3D models are at present difficult to apply for large
areas such as the Mekong Delta that require more detailed data and much more computational
capacity (Soumendra et al., 2010; Dung et al., 2011).  For the aims of this study, it was necessary
to focus on a large part of the delta, as we were interested on the impacts of upstream measures
on downstream river water levels. Given the constraints in data and available model
configurations, we combined the 1D model with a quasi-2D approach as the most suitable
modelling approach. Our modelling results are in line with previous studies applying similar
approaches. Our analysis of the water balance, however, suggests it is recommended to start
investing in better and more comprehensive data availability, as well as computational capacity,
in order to gain a more in-depth understanding of the delta's flood behaviour through additional
2D and 3D modelling.
In this paper the hydraulic impact of upstream dike construction on the flood hazards
downstream the Mekong Delta is assessed through modelling of dike density scenarios in the
flood hydrographs of 2011 and 2013. To do this, the existing Mekong delta one-dimensional
(1D) hydrodynamic model was combined with a quasi-two-dimensional (2D) approach to
explore the change in water dynamics downstream under large-scale high-dike developments
in An Giang and Long Xuyen Quadrangle. Most hydrodynamic studies of the Mekong Delta
limit their assessment of flood risks to the impacts on flood water levels and the spatial extent
of flooding – usually through retrofitting of modelled changes (e.g. dikes and canal network) to



past flood events (e.g. flood levels and flood area data). Whereas good fits are generally
reported between model outputs and recorded water levels, these studies are unable to explain
how the flood volumes are distributed over the delta. To address this, this study quantified the
water balances of the studied flood scenarios and events, to provide insight in the model's
spatial (re) distribution of flood volumes as a result of alterations in dike construction.
**6. Conclusions**
In conclusion, our model results indicate that the extensive construction of high dikes in the
upstream floodplains has only a limited impact on the downstream peak river water levels in
Can Tho due to limited model water inflow. There is, however, a more substantial impact at the
upstream stretches of the river and delta. Our modelling results indicate that dike construction
results in a reduction of floodwater volumes reaching the Long Xuyen Quadrangle in excess of
the retention volume lost through dike construction. Moreover, the model allows extensive
poldering to be associated with only limited increases in river water levels. Limited inflow and
hence the diversion of flood volumes to the Plain of Reeds and Cambodian floodplain can,
however, cannot be validated due to a lack of monitoring data. Future assessments of flood risks
need to be conducted on the scale of the Lower Mekong Basin, with explicit calibration and
validation of the Cambodian floodplain and Plain of Reeds to trace the re-distribution of flood
volumes and water peaks across the delta.
**Acknowledgement**
This research for this paper was funded by NUFFIC/NICHE VNM 104 project, which is co-
funded by the Netherlands Government and Vietnam National University. We would like to
thank the experts in Water Resources Management Group at Wageningen University, the
Netherlands and Center of Water Resources Management in Vietnamese National University





for their valuable comments and supports. In addition, we are grateful to the support of DHI in
providing a license of the Mike 11 model during the study.

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

2                                      (validation)

| Location | Correlation coefficient R² | | | | Nash-Sutcliffe efficiency E | | | |
| --- | --- | --- | --- | --- | --- | --- | --- | --- |
| | WL 2011 | WL 2013 | Q 2011 | Q 2013 | WL 2011 | WL 2013 | Q 2011 | Q 2013 |
| Tan Chau | 0.97 | 0.96 | 0.97 | 0.95 | 0.94 | 0.90 | 0.88 | 0.94 |
| Chau Doc | 0.95 | 0.90 | 0.95 | 0.92 | 0.92 | 0.79 | 0.92 | 0.90 |
| Vam Nao | 0.98 | 0.94 | 0.96 | 0.94 | 0.91 | 0.93 | 0.90 | 0.92 |
| Long Xuyen | 0.96 | 0.93 | - | - | 0.92 | 0.92 | - | - |
| Can Tho | 0.97 | 0.98 | 0.95 | 0.96 | 0.97 | 0.97 | 0.92 | 0.90 |
| Cao Lanh | 0.97 | 0.94 | - | - | 0.93 | 0.94 | - | - |
| My Thuan | 0.97 | 0.94 | 0.89 | 0.91 | 0.94 | 0.83 | 0.80 | 0.86 |
| Xuan To | 0.91 | 0.90 | - | - | 0.85 | 0.87 | - | - |
| Tri Ton | 0.95 | 0.93 | - | - | 0.91 | 0.85 | - | - |
| Tan Hiep | 0.97 | 0.93 | - | - | 0.96 | 0.90 | - | - |
| Phung Hiep | 0.94 | 0.94 | - | - | 0.85 | 0.88 | - | - |

3                *(-) Missing data due to the availability of observed discharge stations.*





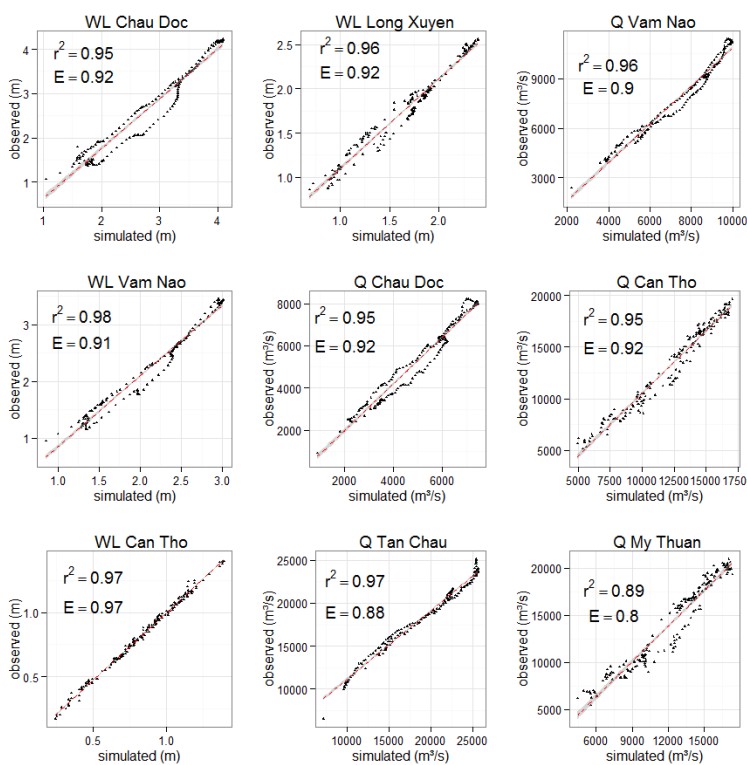

2     Figure 3: Graphs of correlation and Nash-Sutcliffe efficiency of simulated and observed flows

3             in 2011 at representative stations





Table 2: Tidal water levels, in number of hours above different thresholds, observed at My
Thanh and Ben Trai stations during the 2000 and 2011 wet season (July to December).

| Water Level | Numbers of hour at My Thanh | | Numbers of hour at Ben Trai | |
|---|---|---|---|---|
| | 2000 | 2011 | 2000 | 2011 |
| >1.5 m | 95 | 424 | 31 | 102 |
| >1.6 m | 35 | 290 | 8 | 51 |
| >1.7 m | 7 | 198 | 0 | 23 |
| >1.75 m | 3 | 160 | 0 | 12 |
| >1.85 m | 1 | 104 | 0 | 0 |

Table 3: River Water Level Changes and origins at Can Tho for 2000 and 2011

| WL at Can Tho | Model (m) | Observed (m) | Δ (m) | Flood volume of VMD ($10^9$ $m^3$) |
|---|---|---|---|---|
| 2000 | 2.02* | 1.79 | -0.23 | 402 |
| 2011 | 2.10 | 2.15 | +0.05 | 283 |
| Δ (m) | **0.08** | 0.36 | **0.28** | |

*Model outcomes for 2000 are derived by using the observed hydrograph and tidal water levels
of 2000 on to network and floodplain characteristics of 2011.





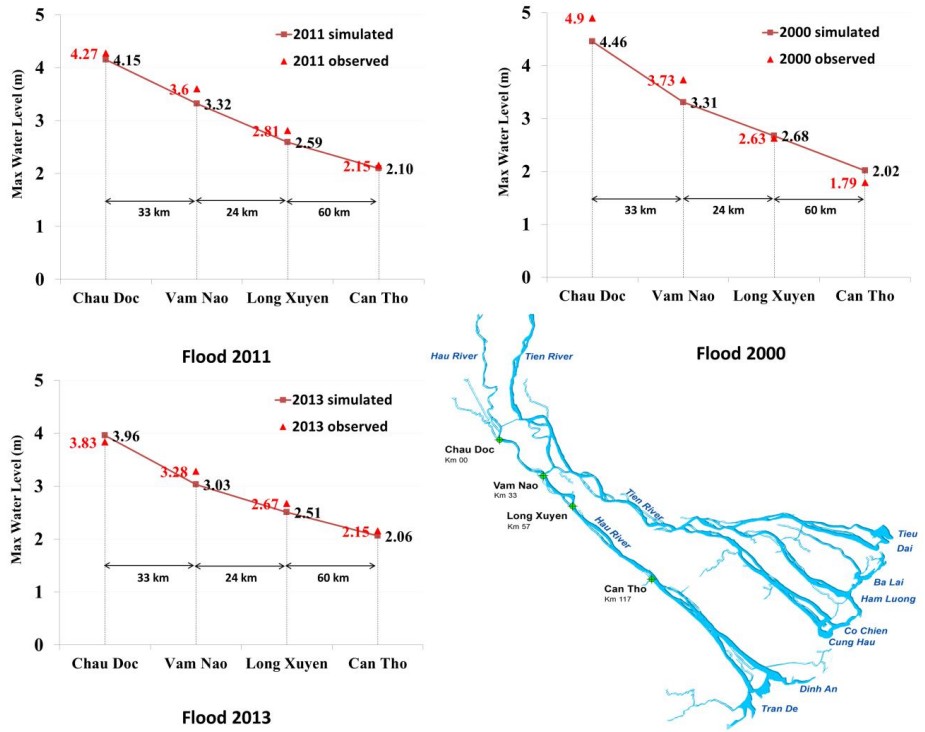

2       Figure 4: Simulated and observed maximum water levels for the 2000, 2011 and 2013 flood

3                           years at 4 different stations along the Hau river.





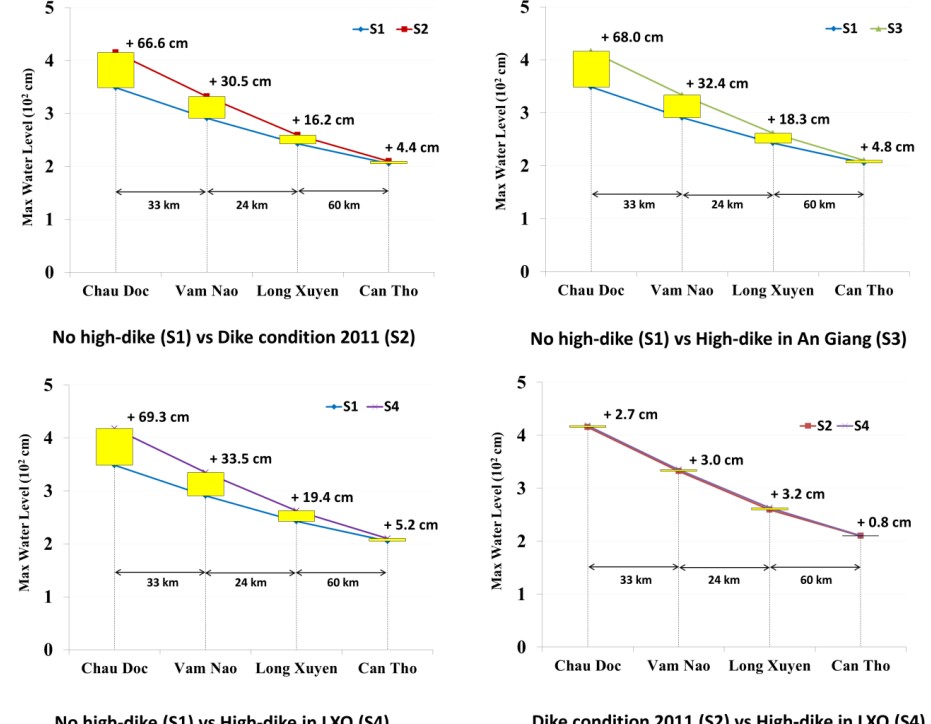

2    Figure 5: Comparisons of the changes in peak water level between scenarios





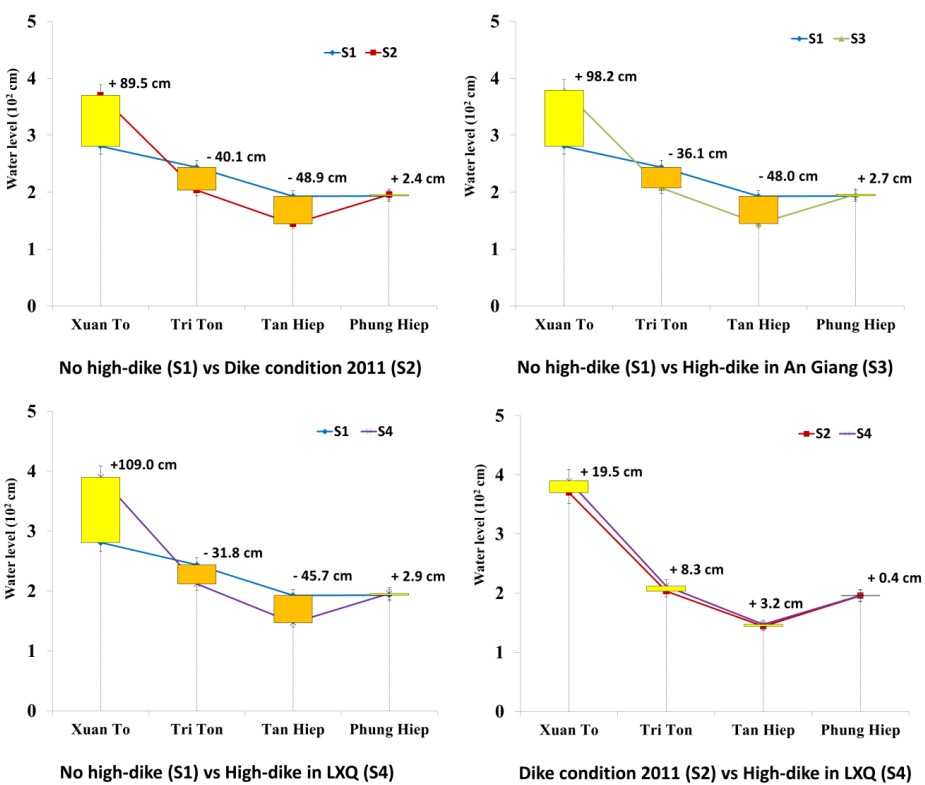

Figure 6: Peak water level changes in the Long Xuyen Quadrangle





1    Table 4: Peak water levels under different dike construction scenarios along the boundary

2    canals of the Long Xuyen Quadrangle

| Scenarios | S1 (m) | S2 (m) | S3 (m) | S4 (m) | S2-S1 (cm) | S3-S1 (cm) | S4-S1 (cm) |
|---|---|---|---|---|---|---|---|
| **Along Vinh Te canal** | | | | | | | |
| (1) Km 0 | 3.40 | 4.18 | 4.20 | 4.22 | 78.00 | 80.70 | 82.20 |
| (2) Km 17 | 3.08 | 3.92 | 3.97 | 4.03 | 84.60 | 89.60 | 95.50 |
| (3) Km 31 | 2.39 | 3.01 | 3.31 | 3.64 | 61.70 | 92.20 | 124.80 |
| (4) Km 42 | 2.77 | 2.95 | 3.01 | 3.61 | 18.80 | 24.00 | 84.20 |
| (5) Km 54 | 2.81 | 2.98 | 3.04 | 3.62 | 17.20 | 22.90 | 81.30 |
| **Along Cai San canal** | | | | | | | |
| (1) Km 0 | 2.36 | 2.31 | 2.33 | 2.34 | -4.30 | -2.80 | -1.80 |
| (2) Km 10 | 2.23 | 1.98 | 2.00 | 2.01 | -25.20 | -23.20 | -21.50 |
| (3) Km 22 | 2.10 | 1.80 | 1.81 | 1.83 | -30.00 | -28.80 | -26.40 |
| (4) Km 33 | 1.99 | 1.53 | 1.54 | 1.56 | -45.80 | -44.90 | -42.50 |
| (5) Km 47 | 1.51 | 1.08 | 1.08 | 1.09 | -42.90 | -42.50 | -41.90 |
| **The canal along the Gulf of Thailand** | | | | | | | |
| (1) Km 0 | 1.11 | 1.14 | 1.05 | 1.02 | 2.50 | -6.70 | -9.40 |
| (2) Km 17 | 1.02 | 1.11 | 1.14 | 1.05 | 9.40 | 11.90 | 2.70 |
| (3) Km 38 | 1.10 | 1.09 | 1.10 | 1.02 | -1.00 | -0.80 | -8.40 |
| (4) Km 56 | 1.35 | 1.06 | 1.04 | 0.98 | -29.20 | -31.60 | -37.20 |
| (5) Km 74 | 1.29 | 0.95 | 0.95 | 0.92 | -34.10 | -34.20 | -36.80 |





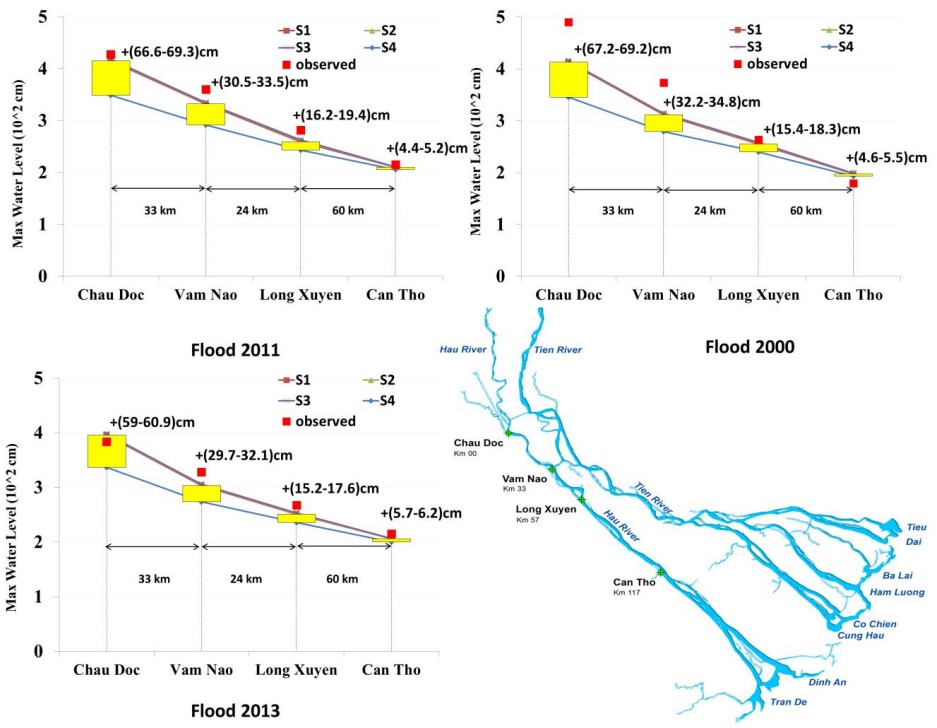

2      Figure 7: Comparison of highest water level changes between scenarios in different floods

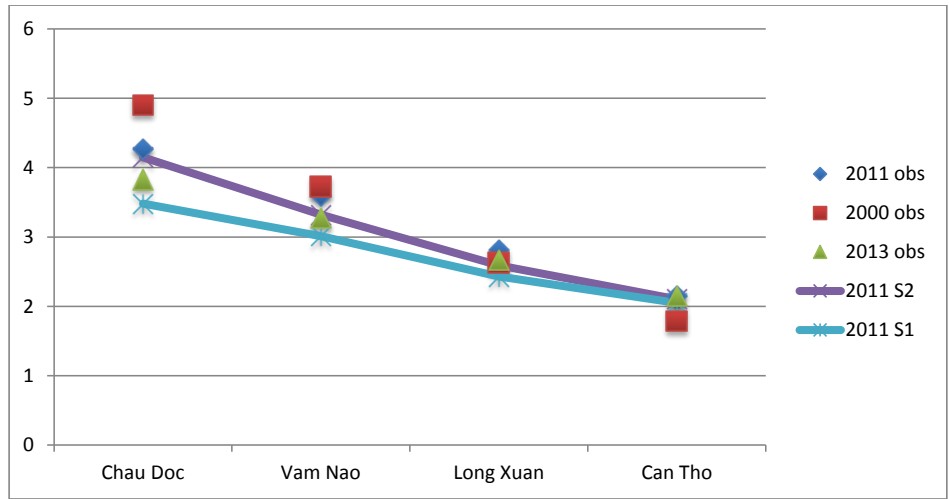

4      Figure 8: Observed and assessed water level along the Hau River





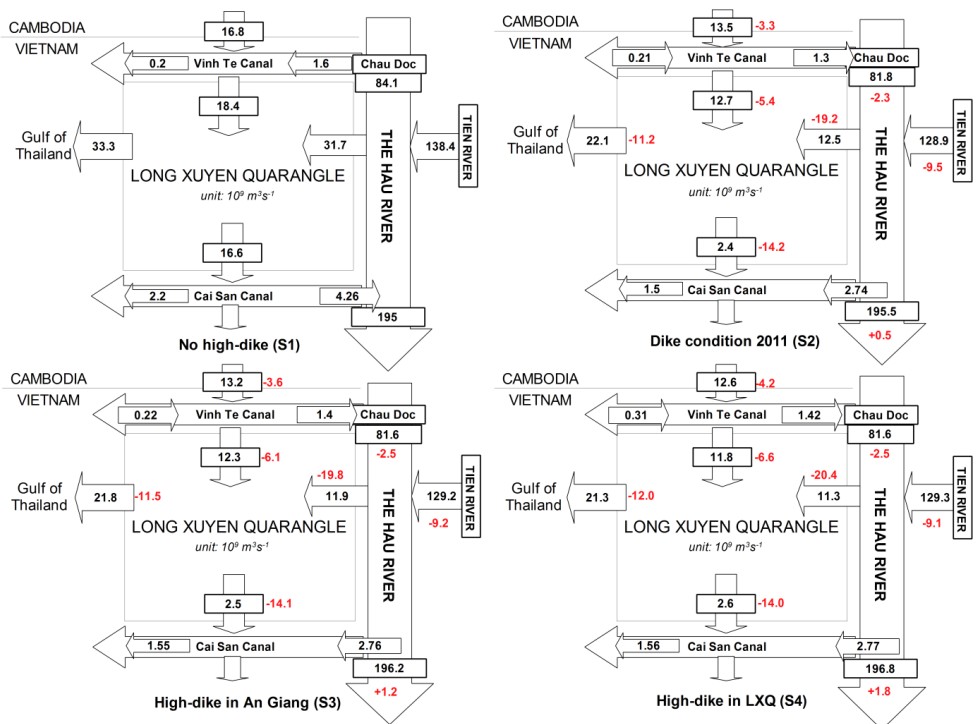

2       Figure 9: Water balance calculation for Long Xuyen Quadrangle under scenarios.  Red

3              number indicate the differences with the S1 scenarios without any high dikes

5       Table 5: Comparison of total inflow to the system

| Scenario | $\sum Qin$ (system) ($10^9$ m$^3$) | $\sum \Delta Qin$ ($10^9$ m$^3$) Compared to the S1 | $\sum \Delta S$ ($10^9$ m$^3$) Water Storage in LXQ |
|---|---|---|---|
| No high-dike (S1) | 239.3 | - | 13 |
| Dike condition 2011 (S2) | 224.2 | -15.1 | 6 |
| High-dike in An Giang (S3) | 224.0 | -15.3 | 4 |
| High-dike in LXQ (S4) | 223.5 | -15.8 | 0 |



1 **ANNEX**

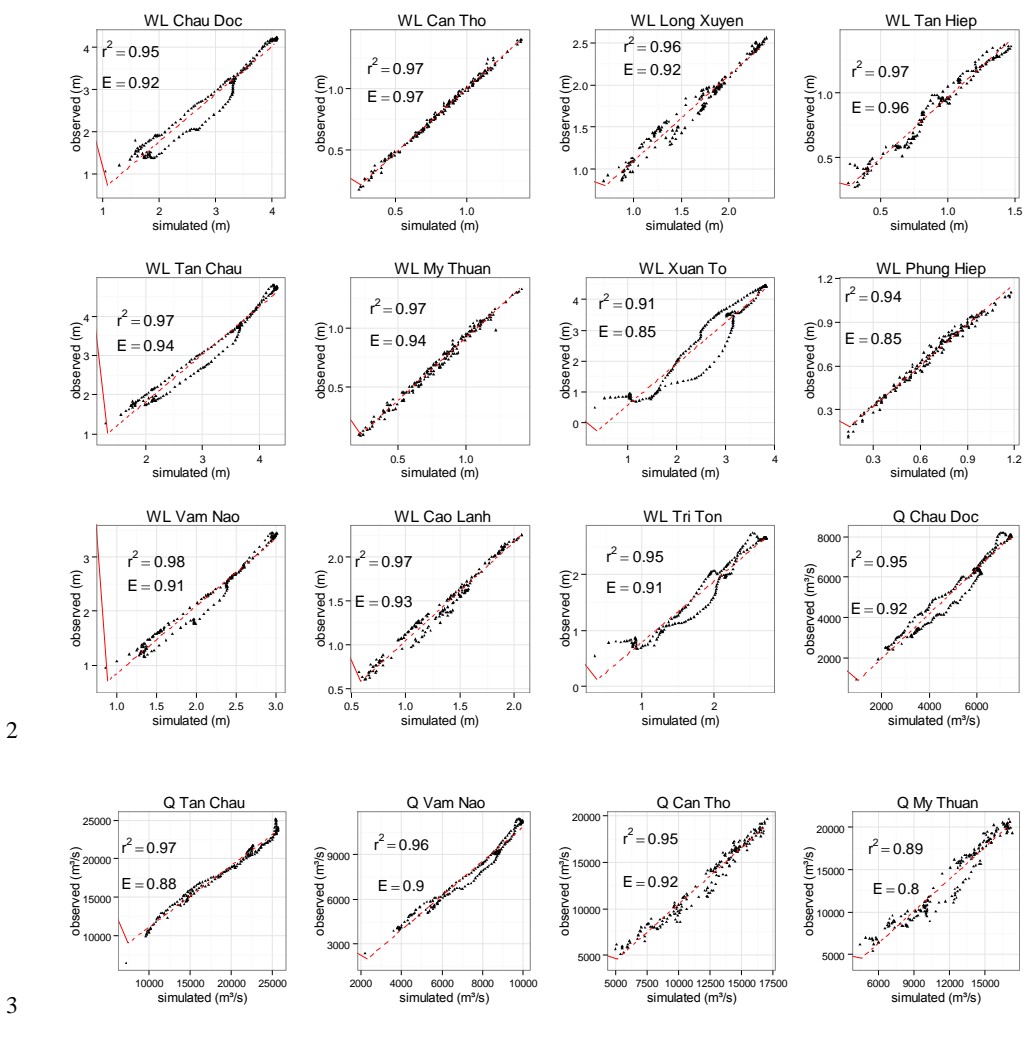

