# Peer review of "Assessing impacts of dike construction on the flood"

_Hydrology and Earth System Sciences, 2017_

## Referee Comment (RC1) · Anonymous Referee #1 · 26 Jun 2017

The authors used a hydrology model to evaluate the relationship between dike construction and the hydrography during the flood season in the Vietnamese Mekong Delta. In general, this study is interesting and important, considering the frequency of flood events and high-density dike constructions. However, the impact of artificial construction on flood dynamic obviously has been extensively studied in many previous literature, so I would not say this is a novel study although the numerical modeling method is rarely seen. There are several issues that needed to be addressed before the paper can be accepted in HESS. I recommend a major revision with further review by the editors and referees.

Here are some comments in the manuscript:

P1, Ln19: I expect the authors to explain and define some technical terms/words at

the first time in the paper, such as the high-dike and semi-dike. In addition, please use the consistent word throughout the paper, for example, correct the August-dike to semi-dile.

P1, Ln22-23: This sentence needs to be rephrased: . . . is assessed through the flood hydrographs modeling under different dike density scenarios in 2011 and 2013.

P2 Ln2: What is a Quadrangle? I don't think it is a right word for hydrology study. Try to use the watershed name.

P2, Ln18-20: I don't suggest to write the future work in the abstract, since it is not part of the authors' work reported in this paper. Also, the authors mentioned that the historical monitoring data are absent, so it is actually difficult or even impossible to do the future assessments.

P3, Ln2-8: The information of economic and food production is too detailed within this paper. Try to shorten this part.

P3, Ln9-17: The economic cost and loss are not related to the scientific question in this paper.

P5, Ln1-3: The authors used one point observation to demonstrate a clear correlation between the dike construction and water level. There is no clear evidence showing the cause and effect between the dike construction and water level. In addition, I don't suggest to write one data point in the introduction part.

P5, Ln9-24: I recommend the authors to cite the reference right after each reason of flood risk (Ln10-11), instead of explaining each reference separately in detailed. The other reasons associated with the flood risks are not strongly related to the scientific questions discussed in this paper. And, are these studies focus on the same study site (Vietnamese Mekong Delta) as well?

P6 Ln22-23: I suggest the authors highlight the gap of modelling approach within this manuscript. The previous studies of modelling approaches and applications should be

addressed in the introduction as well.

P8 Ln4: What is a.m.s.l?

P9 Ln3-5: Again, move these explanations forward to help readers understand their meanings.

P8 Methodology section: I think a more detailed introduction of the hydrologic model and software are necessary, including the governing equations and physics used in the model, their applications, pros and cons, etc. The authors can't just cite the references.

P11 Ln5-7: How did you select these Manning coefficients? Any references? And, try to use a clearer word rather than "global" to avoid misunderstanding.

P14 Ln19: I'm confused about the Q-Q plots in figure 3. What does each point represent? Are they daily simulated and observed streamflows? If yes, the authors should consider the time-series plot for the streamflow results.

P15 Ln12-13: The x-axis of figure 4 should be the distance instead of the sites. I expect the explanation of underestimated simulated streamflow to be right after the figures. P15 Ln20: Should be 2011?

P17 Ln5-6, Ln 15: A general comment: The authors should use more quantitative criteria to demonstrate either the difference or similarity between different scenarios. Statistical measures are highly desired.

P17 Ln 20-23: The authors should provide a more detailed explanation of the "hinge response". A modified hydrograph as forcing should be presented in the paper as well to help readers understand the "hinge response".

P21 Discussion section: In general, I think the discussion section is too long and verbose. The discussion must be shortened with clearer statements for each analysis. The authors can also try to reconstruct the discussions with results section to help the readers better understand the highlighted study results.

P26: Again, a reconstruction of discussion and conclusion sections is needed for this paper. I strongly recommend the authors to use bullets to clearly state the major findings of this study in the conclusion part.

Tables and figures should be listed separately. Why did you list the peak water levels in Table 4 instead of plotting in the figures? Probably try to plot the peak water levels under different dike construction as well unless any other reasons.

---

## Referee Comment (RC2) · Anonymous Referee #2 · 3 Jul 2017

This paper is about assessing the impacts of dike construction on the flood dynamics in the Mekong Delta. This is an interesting topic, because the number of floods in this region is increasing. However, several aspects in this paper have to be thoroughly revised because it can be published. This is explained below. Therefore, I recommend a major revision before this paper can be published.

• Readability; The paper is not to-the-point. This paper is about a 1D-hydrodynamic model that has been calibrated and validated for floods in 2011 and 2013. However, the introduction in Chapter 1 is very long and contains many aspects that are not relevant for this study. The same holds for Chapter 2. Also the discussion in Chapter 5 is much too long and should be made to-the-point. Please rewrite Chapters 1 to 6 in a to-the-point way, so that the number of pages will reduce significantly. • Description

of model set-up; A crucial aspect is the flooding in cross-sectional direction in this 1D model. A quasi-2D approach is applied for the flood plains. This is explained very briefly and should be explained in detail, because it has a large impact on the model results. • Lack of validation data; In the abstract and in the conclusions is stated that there is a lack of validation data. However, in Section 3.2 is stated that hourly discharge and water levels are available at 15 locations, of which four locations are even in flood plans. This is a nice validation set. So, there seems to be no lack of validation data. • Presentation of model results; One expects figures with time series of water levels and discharges that contain both numerical results and measurements. However, such plots are missing. Therefore, a reader does not have any insight whether the time behavior of the floods is simulated accurately. Instead, correlation numbers and maximum high water are presented in the figures. However, this is of secondary importance. The authors are strongly advised to add several time history figures with computed and measured results. • Description of dykes; Please clarify the differences between the dyke types (semi-dyke, August dike, high dyke) that are mentioned in this paper. • Hinge response; What is a hinge response? Please clarify. • What is new in this paper? In the discussion (P. 21) is stated that the results are consistent with earlier studies with 1D hydrodynamic models. What is new in this paper? Please add the references of the other studies. • Validation for 2000 flood. Suddenly in the paper the authors start with a validation of the 2000 flood. The results are not very accurate because the geometry of the Mekong Delta was somewhat different in 2000. What is the purpose of this validation? Should this be left out? • Accuracy of model results. In the discussion (P. 25) is stated that the model results are in line with other studies with 1D model, but that that 2D (and possible) 3D modeling is required for an in-depth understanding of the flood behavior in the Mekong Delta. In other words, do the authors conclude that 1D modeling with a quasi-2D approach for flooding is not suitable for this?

Please also note the supplement to this comment:

https://www.hydrol-earth-syst-sci-discuss.net/hess-2017-141/hess-2017-141-RC2-supplement.pdf

---

## Author Comment (AC1) · 21 Aug 2017

We highly appreciate Reviewer#1 for the dedicated reviews and valuable comments on the manuscript. Please find below our details responses and corresponding revisions. We have uploaded our responses as a supplement. Please also note the supplement to this comment.

Please also note the supplement to this comment:
https://www.hydrol-earth-syst-sci-discuss.net/hess-2017-141/hess-2017-141-AC1-supplement.pdf
* * *
[Figure]

141, 2017.

**Supplement:**

**Dung Duc Tran et al. Assessing impacts of dike construction on the flood dynamics in the Mekong Delta**

Responses to Reviewer#1 comments

By Dung Duc Tran

dung.ductran@wur.nl or dungtranducvn@yahoo.com

**General comment**: The authors used a hydrology model to evaluate the relationship between dike construction and the hydrography during the flood season in the Vietnamese Mekong Delta. In general, this study is interesting and important, considering the frequency of flood events and high-density dike constructions. However, the impact of artificial construction on flood dynamic obviously has been extensively studied in many previous literature, so I would not say this is a novel study although the numerical modeling method is rarely seen. There are several issues that needed to be addressed before the paper can be accepted in HESS. I recommend a major revision with further review by the editors and referees.

**Response**: We highly appreciate Reviewer#1 for the dedicated reviews and valuable comments on the manuscript. Please find below our details responses and corresponding revisions.

**1. Comment P1, Ln19**: I expect the authors to explain and define some technical terms/words at the first time in the paper, such as the high-dike and semi-dike. In addition, please use the consistent word throughout the paper, for example, correct the August-dike to semi-dike.

**Response:** We agree with Rewiewer#1 to define some key terms (i.e. high-dike, semi-dike) at the beginning of the paper. The high-dikes we considered in the study are the closed polders/compartments used mainly for protecting triple-rice production. We add short text to describe dike types in the Abstract and further elaborate on these terms in the main text.

Added text to the Abstract: "*for triple-rice production*"

Regarding the semi-dike, we define it in the Introduction section of the manuscript only because this dike is not considered as a factor causing flood risk downstream. However, we will replace the "semi-dike" term with "low-dike" term. This term helps readers understand the meaning easily in reference to our use of "high-dike". In addition, the "low-dike" term will be used throughout the manuscript to ensure the consistency.

**2. Comment P1, Ln22-23**: This sentence needs to be rephrased:...is assessed through the flood hydrographs modeling under different dike density scenarios in 2011 and 2013.

**Response:** We will rephrase the sentence as Reviewer#1's comment.

Rephrased text: "*This paper assesses the hydraulic impact of upstream dike construction on the flood hazards downstream the Mekong Delta. To do this, the existing Mekong delta one-dimensional (1D) hydrodynamic model combined with a quasi-two dimensional (2D) approach was first calibrated and validated with floods in 2011 and 2013, then used to explore the water dynamics downstream under extensive high-dike developments in An Giang and Long Xuyen Quadrangle*"

**3. Comment P2 Ln2**: What is a Quadrangle? I don't think it is a right word for hydrology study. Try to use the watershed name.

**Response:** We agree with the reviewer that using a watershed name is better in a hydrology study. But in our case, "Long Xuyen Quadrangle" term is widely used in literature (Hung et al., 2012, 2014a, 2014b, Manh et al., 2014a, 2015; Mekong Delta Plan, 2013), referring to one of the two geographical floodplains in the Vietnamese Mekong Delta (the other floodplain is Plain of Reeds). In addition, Figure 1 shows the geographical extent of the Long Xuyen Quadrangle in the manuscript which helps readers know the location of this floodplain. We also described the Long Xuyen Quadrangle in the Introduction section.

**4. Comment P2, Ln18-20**: I don't suggest to write the future work in the abstract, since it is not part of the authors' work reported in this paper. Also, the authors mentioned that the historical monitoring data are absent, so it is actually difficult or even impossible to do the future assessments.

**Response:** We fully agree with Reviewer#1. These sentences will be removed from the Abstract.

**5. Comment P3, Ln2-8**: The information of economic and food production is too detailed within this paper. Try to shorten this part.

**Response:** We agree and will shorten this part in the revised manuscript. In particular we will merge two paragraphs about flood benefits and damages into one and remove some unnecessary details.

**6. Comment P3, Ln9-17**: The economic cost and loss are not related to the scientific question in this paper.

**Response:** We agree with the reviewer to shorten the content in the revised manuscript.

**7. Comment P5, Ln1-3**: The authors used one point observation to demonstrate a clear correlation between the dike construction and water level. There is no clear evidence showing the cause and effect between the dike construction and water level. In addition, I don't suggest to write one data point in the introduction part.

**Response:** We agree with Reviewer#1 that one point observation could not be used to demonstrate a clear correlation between the dike construction and the increase in water levels. We use this example to highlight a public concern about the flood risk downstream caused by upstream dike construction. It was confused for readers when we used the "suggest there is a clear correlation" phrase in the manuscript. We will rephrase this sentence as one of the evidence/example indicating flood risks downstream which would be potentially exacerbated by the large-scale high-dike constructions. This evidence is also presented in a recent study by Triet et al. (2017).

Text would be changed as follows: *"For example, the water level observed in 2011 at the upstream station of Tan Chau was 0.63 m lower than in 2000 (4.27m and 4.90 m), whereas the water levels observed in Can Tho were 0.36 m higher in 2011 than in 2000 (2.15 m and 1.79 m, respectively). This thus indicates a correlation between the increase in high dike construction in the floodplains and the increase of water level and flood risks downstream in and around the city of Can Tho".*

**8. Comment P5, Ln9-24**: I recommend the authors to cite the reference right after each reason of flood risk (Ln10-11), instead of explaining each reference separately in detailed. The other reasons associated with the flood risks are not strongly related to the scientific questions discussed in this paper. And, are these studies focus on the same study site (Vietnamese Mekong Delta) as well?

**Response:** We thank Reviewer#1 for this useful suggestion. We will rewrite this paragraph to better connect references to different reasons of flood risks. We will also revise the text to link the information with the objective of the paper. We will also add text to inform the study site (VMD) of the referenced papers.

Revised text: *"Several studies have concluded that flood risks in the VMD delta have increased over the last decades due to a number of reasons: climate change, sea level rise, hydropower development, land subsidence, and local rainfall. In the complex hydrodynamic context of the delta, flood risks are assessed in various forms by a variety of authors: flood extent and duration, flood depth or floodwater level, or river water levels. Wassmann et al. (2004) used a hydraulic model to conclude the increase in water level in the delta caused by sea level rise under changing climate. Fujihara et al. (2015) conducted a study to identify the possible impacts of the runoff upstream, sea level rise, and land subsidence on the floodwater level rise. Lauri et al., (2012) and Hoang et al., (2016) assessed the impacts of projected climate change and reservoir operation on the future changes in the Mekong River hydrology. Some other studies showed the impacts of climate change and sea level rises on flood propagation, flood inundation and sediment transport (Apel et al., 2012; Hung, 2012b; Quang et al., 2012; Manh et al., 2014)."*

**9. Comment P6 Ln22-23**: I suggest the authors highlight the gap of modelling approach within this manuscript. The previous studies of modelling approaches and applications should be addressed in the introduction as well.

**Response:** We will highlight the important gaps of the modelling approach, namely the lack of mechanistic understanding about impacts of upstream high-dike development on downstream flood hazards, and the missing quantifications of the water balance. We will add text to emphasize these two gaps in the revised manuscript.

Added text to the revised manuscript: *"Despite rapid development of high-dike system for triple rice production in the upper Mekong Delta, few modeling studies assessed the resulting implication of such developments on floodwater regimes. As a result, understandings about the changing regimes and their driving factors remain very limited. Additionally, while previous studies focused strongly on variation of peak water levels based on monitoring data or model results, no study analyzed changes in the floodwater balance. Since the water balance analyses are important to understand where floodwater is distributed under different dike scenarios, we use a 1D-quasi2D modelling approach to address this knowledge gap. This approach allows to test the hypothesis that large-scale high-dike constructions would reduce the flood retention capacity of the floodplains and increase water levels (and associated flood risks) along rivers downstream."*

**10. Comment P8 Ln4**: What is a.m.s.l?

**Response:** We will clarify the term of "a.m.s.l" as "*above mean sea level*".

**11. Comment P9 Ln3-5**: Again, move these explanations forward to help readers understand their meanings.

**Response:** We agree with Reviewer#1 to move these explanations forward to help readers understand the characteristics of low dikes and high dikes. In the revised manuscript, these explanations will be moved forward in the first paragraph of Section 2 (**Study area**).

**12. Comment P8 Methodology section**: I think a more detailed introduction of the hydrologic model and software are necessary, including the governing equations and physics used in the model, their applications, pros and cons, etc. The authors can't just cite the references.

**Response:** We will add a brief introduction of the hydrologic model and software. In the revised manuscript, we will also present detailed introduction in the Supplement.

Added text to the revised manuscript: "*The one-dimensional hydrodynamic model based on Mike 11 software developed by the Danish Hydraulic Institute (DHI). The model is an implicit finite difference model for one-dimensional unsteady flow computation. In addition, it could be applied to quasi-two dimensional (quasi-2D) flow simulation to perform detailed modelling of rivers, including special treatment of floodplains, road overtopping, culverts, gate openings and weirs (Doulgeris et al., 2012). The model is capable of using kinematic, diffusive or fully dynamic, vertically integrated equations of conservation of continuity and momentum (the Saint-Venant equations) for solving complex flow and mass transport problems* (Patro et al., 2009; Dung et al., 2011; Manh et al., 2014b)*. Therefore, it was used to represent the river network and its floodplains for the Mekong Delta. Detailed information about the equations and computational components is shown in the Supplement 4.*"

Text added in the Supplement:

*The Saint-Venant equations are formulated as follows* (DHI, 2011)*:*

*Continuity equation:*

$$\frac{\partial Q}{\partial t} + \frac{\partial A}{\partial t} = q$$

*Momentum equation:*

$$\frac{\partial Q}{\partial t} + \frac{\partial \left(\frac{\propto Q^2}{A}\right)}{\partial x} + gA\frac{\partial h}{\partial x} + \frac{gQ|Q|}{C^2 AR} = 0$$

*Where*

*Q-discharge [m$^3$/s]*

*A-flow area [m$^2$]*

*q-the lateral inflow [m$^2$/s]*

*h-stage above datum [m]*

*C-Chezy resistance coefficient [m$^{1/2}$/s]*

*R-hydraulic or resistance radius [m]*

∝-*momentum distribution coefficient*

**13. Comment P11 Ln5-7**: How did you select these Manning coefficients? Any references? And, try to use a clearer word rather than "global" to avoid misunderstanding.

**Response:** We agree to explain our selection of the Manning Coefficients in the revised manuscript and will add references. Beside, "global" term is replaced with "generic" and the sentence will be clarified to explain our input for a coefficient for the whole rivers in the network.

Explained how to select the Manning Coefficients and references: "*The initial river roughness were estimated based upon the range of roughness values corresponding to types of rivers and canals provided by various publications (Chow, 1959; Fabio et al., 2010; Dung et al., 2011)*".

The text will be rewritten as follows: "*These roughness numbers are represented as manning coefficients in the model. Referring to Chow (1959), we first set generic manning coefficients of 0.02 (irrigation channel, straight, in hard-packed smooth sand), 0.025 (earth channel exacerbated in alluvial silt soil, with deposits of sand on bottom and growth of grass), and 0.033 (natural channel, somewhat irregular side slopes, very little variation in cross section) respectively for the whole river branches for the three initial runs to identify the changes in water levels and discharge in the main rivers*"

**14. Comment P14 Ln19**: I'm confused about the Q-Q plots in figure 3. What does each point represent? Are they daily simulated and observed streamflow? If yes, the authors should consider the time-series plot for the streamflow results.

**Response:** Each point in the Q-Q plot in Figure 3 represents the correlation between daily simulated and observed streamflow, based on Correlation Coefficient ($R^2$) and Nash-Sutcliffe Efficiency (NSE).We agree with Reviewer#1 to present some time-series plots in the manuscript instead of the Q-Q plots. Another reason is that Table 1 already shows the NSE and $R^2$. All the Q-Q plots will be presented only in the Supplement. We will also add all time-series plots to the Supplement. Figure 3 will be captioned as "Time series of simulated and observed flows in 2011 at representative stations".

[Figure]

*Figure 3: Time-series of simulated and observed flows in 2011 at representative stations*

In Supplement:

[Figure]

*Figure S2: Time-series of daily simulated and observed flows in 2011 at all stations used for model calibration*

[Figure]

*Figure S3: Time-series of daily simulated and observed flows in 2013 at all stations used for model calibration*

**15. Comment P15 Ln12-13**: The x-axis of figure 4 should be the distance instead of the sites. I expect the explanation of underestimated simulated streamflow to be right after the figures.

**Response:** In the submitted manuscript, we used the 4 sites instead the distance shown in the axis because these are important points measured along the Hau River. For example, Chau Doc

and Can Tho are two points we exlore the difference in measured water levels between the floods of 2000 and 2011. In addition, the x-axis of Figure 4 also shows the distance of 4 sites. We will add an explanation of underestimated simulated streamflow from Figure 4 (below).

We will add an explanation to Figure 4:

[Figure]

Figure 4: Simulated and observed maximum water levels for the 2000, 2011 and 2013 flood years at 4 different stations along the Hau River.

Added text right after the figure: *"The figure shows a good fitness of simulated and observed peak water levels in floods of 2011(calibration) and 2013 (validation). In the flood 2000, the fitness is low due to the significant changes in physical topography such as river cross-sections between the model setup of 2011 and the measured data in 2000."*

**16. Comment P15 Ln20**: Should be 2011?

**Response:** We will correct the number to "2011".

**17. Comment P17 Ln5-6, Ln 15**: A general comment: The authors should use more quantitative criteria to demonstrate either the difference or similarity between different scenarios. Statistical measures are highly desired.

**Response:** We thank for the comment of Reviewer#1. We will add some statistical calculations to the Supplement to present the difference and similarity between water levels in different dike scenarios. In addition, we will explain whether there are significant differences in the water levels or not in the text of the revised manuscript.

In Supplement:

Table S1: Paired sample test for water level time series along the Hau River in 2011

**Paired Sample Test for water level (m) time series at Chau Doc**

| Scenario and Difference | N | Mean | Peak | Peak Time | Std. Deviation | 95% Confidence interval of the difference | | t-value | df | p-value |
|---|---|---|---|---|---|---|---|---|---|---|
| | | | | | | Lower | Upper | | | |
| S1 | 4393 | 2.567 | 3.486 | 12/10/2011 | 0.669 | | | | | |
| S2 | 4393 | 2.908 | 4.152 | 12/10/2011 | 0.882 | | | | | |
| S3 | 4393 | 2.912 | 4.166 | 12/10/2011 | 0.885 | | | | | |
| S4 | 4393 | 2.920 | 4.179 | 12/10/2011 | 0.890 | | | | | |
| Pair S1-S2 | | | | | | -0.374 | -0.308 | -20.415 | 8188 | 0.000 |
| Pair S1-S3 | | | | | | -0.377 | -0.311 | -20.569 | 8175 | 0.000 |
| Pair S1-S4 | | | | | | -0.385 | -0.319 | -20.968 | 8153 | 0.000 |
| **Paired Sample Test for water level (m) time series at Vam Nao** | | | | | | | | | | |
| S1 | 4393 | 1.937 | 2.664 | 13/10/2011 | 0.521 | | | | | |
| S2 | 4393 | 2.030 | 2.943 | 26/10/2011 | 0.583 | | | | | |
| S3 | 4393 | 2.035 | 2.963 | 26/10/2011 | 0.588 | | | | | |
| S4 | 4393 | 2.040 | 2.975 | 26/10/2011 | 0.593 | | | | | |
| Pair S1-S2 | | | | | | -0.116 | -0.070 | -7.914 | 8674 | 0.000 |
| Pair S1-S3 | | | | | | -0.122 | -0.075 | -8.304 | 8656 | 0.000 |
| Pair S1-S4 | | | | | | -0.127 | -0.081 | -8.726 | 8640 | 0.000 |
| **Paired Sample Test for water level (m) time series at Long Xuyen** | | | | | | | | | | |
| S1 | 4393 | 1.654 | 2.431 | 27/10/2011 | 0.499 | | | | | |
| S2 | 4393 | 1.653 | 2.593 | 27/10/2011 | 0.509 | | | | | |
| S3 | 4393 | 1.658 | 2.614 | 26/10/2011 | 0.514 | | | | | |
| S4 | 4393 | 1.664 | 2.625 | 26/10/2011 | 0.519 | | | | | |
| Pair S1-S2 | | | | | | -0.020 | 0.022 | 0.083 | 8780 | 0.934 |
| Pair S1-S3 | | | | | | -0.025 | 0.017 | -0.370 | 8776 | 0.711 |
| Pair S1-S4 | | | | | | -0.031 | 0.012 | -0.862 | 8771 | 0.389 |
| **Paired Sample Test for water level (m) time series at Can Tho** | | | | | | | | | | |
| S1 | 4393 | 0.843 | 2.054 | 27/10/2011 | 0.480 | | | | | |
| S2 | 4393 | 0.829 | 2.098 | 27/10/2011 | 0.499 | | | | | |
| S3 | 4393 | 0.830 | 2.102 | 27/10/2011 | 0.499 | | | | | |
| S4 | 4393 | 0.832 | 2.106 | 27/10/2011 | 0.500 | | | | | |
| Pair S1-S2 | | | | | | -0.006 | 0.035 | 1.368 | 8771 | 0.172 |
| Pair S1-S3 | | | | | | -0.008 | 0.033 | 1.197 | 8770 | 0.231 |
| Pair S1-S4 | | | | | | -0.010 | 0.031 | 1.008 | 8770 | 0.314 |

**18. Comment P17 Ln 20-23**: The authors should provide a more detailed explanation of the "hinge response". A modified hydrograph as forcing should be presented in the paper as well to help readers understand the "hinge response".

**Response:** Based on the responses of both reviewers we have come to the conclusion that the use of the term "hinge response" is confusing. Therefore in the revised version of the paper, we will not use the term "hinge response" anymore. We will also rename this section in chapter 4

**19. Comment P21 Discussion section**: In general, I think the discussion section is too long and verbose. The discussion must be shortened with clearer statements for each analysis. The authors can also try to reconstruct the discussions with results section to help the readers better understand the highlighted study results.

**Response:** We will shorten the Discussion section to help readers have better understanding about the significance of the study results.

**20. Comment P26**: Again, a reconstruction of discussion and conclusion sections is needed for this paper. I strongly recommend the authors to use bullets to clearly state the major findings of this study in the conclusion part. Tables and figures should be listed separately. Why did you list the peak water levels in Table 4 instead of plotting in the figures? Probably try to plot the peak water levels under different dike construction as well unless any other reasons.

**Response:** We will substantially revise and tighten the structure of the discussion section and create better links to the main results. We will also rewrite the Conclusion section by using bullets to state the major findings. Regarding Table 4, we would like to keep it in the revised manuscript because we want to show the detailed differences between peak floodwater levels in different points on canals of the floodplain.

**References**

DHI, 2011. A Modelling System for Rivers and Channels. DHI Software Licence Agreement, Danish Hydraulic Institute.

Doulgeris, C., Georgiou, P., Papadimos, D., Papamichail, D., 2012. Ecosystem approach to water resources management using the MIKE 11 modeling system in the Strymonas River and Lake Kerkini. Journal of Environmental Management 94, 132–143. doi:10.1016/j.jenvman.2011.06.023

Dung, N.V., Merz, B., BĂ¡rdossy, A., Thang, T.D., Apel, H., 2011. Multi-objective automatic calibration of hydrodynamic models utilizing inundation maps and gauge data. Hydrology and Earth System Sciences 15, 1339–1354. doi:10.5194/hess-15-1339-2011

Duong, V.H.T., Trinh Cong, V., Franz, N., Peter, O., Nguyen Trung, N., 2014. Land use based flood hazards analysis for the mekong delta, in: Proceedings of the 19 Th IAHR - APD Congress 2014, Hanoi , Vietnam. Presented at the IAHR, Ha Noi, Vietnam. doi:10.13140/2.1.5153.9842

Fujihara, Y., Hoshikawa, K., Fujii, H., Kotera, A., Nagano, T., Yokoyama, S., 2015. Analysis and attribution of trends in water levels in the Vietnamese Mekong Delta. Hydrological Processes n/a-n/a. doi:10.1002/hyp.10642

Hoang, L.P., Lauri, H., Kummu, M., Koponen, J., van Vliet, M.T.H., Supit, I., Leemans, R., Kabat, P., Ludwig, F., 2016. Mekong River flow and hydrological extremes under climate change. Hydrology and Earth System Sciences 20, 3027–3041. doi:10.5194/hess-20-3027-2016

Hung, N.N., Delgado, J.M., Güntner, A., Merz, B., Bárdossy, A., Apel, H., 2014a. Sedimentation in the floodplains of the Mekong Delta, Vietnam Part II: deposition and erosion. Hydrological Processes 28, 3145–3160. doi:10.1002/hyp.9855

Hung, N.N., Delgado, J.M., Güntner, A., Merz, B., Bárdossy, A., Apel, H., 2014b. Sedimentation in the floodplains of the Mekong Delta, Vietnam. Part I: suspended sediment dynamics. Hydrological Processes 28, 3132–3144. doi:10.1002/hyp.9856

Hung, N.N., Delgado, J.M., Tri, V.K., Hung, L.M., Merz, B., Bárdossy, A., Apel, H., 2012. Floodplain hydrology of the Mekong Delta, Vietnam. Hydrological Processes 26, 674–686. doi:10.1002/hyp.8183

Lauri, H., de Moel, H., Ward, P.J., Räsänen, T.A., Keskinen, M., Kummu, M., 2012. Future changes in Mekong River hydrology: impact of climate change and reservoir operation on discharge. Hydrology and Earth System Sciences 16, 4603–4619. doi:10.5194/hess-16-4603-2012

Manh, N.V., Dung, N.V., Hung, N.N., Kummu, M., Merz, B., Apel, H., 2015. Future sediment dynamics in the Mekong Delta floodplains: Impacts of hydropower development, climate change and sea level rise. Global and Planetary Change 127, 22–33. doi:10.1016/j.gloplacha.2015.01.001

Manh, N.V., Dung, N.V., Hung, N.N., Merz, B., Apel, H., 2014a. Large-scale suspended sediment transport and sediment deposition in the Mekong Delta. Hydrology and Earth System Sciences 18, 3033–3053. doi:10.5194/hess-18-3033-2014

Manh, N.V., Dung, N.V., Hung, N.N., Merz, B., Apel, H., 2014b. Large-scale suspended sediment transport and sediment deposition in the Mekong Delta. Hydrology and Earth System Sciences 18, 3033–3053. doi:10.5194/hess-18-3033-2014

Marchand, M., Pham Quang, D., Le, T., 2014. Mekong Delta: Living with water, but for how long? Built Environment 40(2), 230–243.

Mekong Delta Plan, 2013. Kingdom of the Netherlands & The Socialist Republic of Vietnam. Report.

Patro, S., Chatterjee, C., Mohanty, S., Singh, R., Raghuwanshi, N.S., 2009. Flood inundation modeling using MIKE FLOOD and remote sensing data. J Indian Soc Remote Sens 37, 107–118. doi:10.1007/s12524-009-0002-1

Tri, V. P. D., Trung, N.H., Tuu, N.T., 2012. Flow dynamics in the Long Xuyen Quadrangle under the impacts of full-dyke systems and sea level rise. VNU Journal of Science Earth Science 28.

Tri, Van Pham Dang, Trung, N.H., Tuu, N.T., 2012. Flow dynamics in the Long Xuyen Quadrangle under the impacts of full-dyke systems and sea level rise. VNU Journal of Science Earth Science 28.

Triet, N.V.K., Dung, N.V., Fujii, H., Kummu, M., Merz, B., Apel, H., 2017. Has dyke development in the Vietnamese Mekong Delta shifted flood hazard downstream? Hydrology and Earth System Sciences 21, 3991–4010. doi:10.5194/hess-21-3991-2017

Wassmann, R., Hien, N., Hoanh, C., Tuong, T.P., 2004. Sea Level Rise Affecting the Vietnamese Mekong Delta: Water Elevation in the Flood Season and Implications for Rice Production. Climatic Change 66, 89–107. doi:10.1023/B:CLIM.0000043144.69736.b7

---

## Author Comment (AC2) · 21 Aug 2017

We highly appreciate Reviewer#2 for the dedicated reviews and valuable comments on the manuscript. We will address all your comments and substantially revise the manuscript accordingly. Our revisions are described in detail below. We have uploaded our responses as a supplement.

Please also note the supplement to this comment: https://www.hydrol-earth-syst-sci-discuss.net/hess-2017-141/hess-2017-141-AC2-supplement.pdf
* * *
141, 2017.

**Supplement:**

**Dung Duc Tran et al. Assessing impacts of dike construction on the flood dynamics in the Mekong Delta**

Responses to Reviewer#2 comments

By Dung Duc Tran

dung.ductran@wur.nl or dungtranducvn@yahoo.com

**General comment**: This paper is about assessing the impacts of dike construction on the flood dynamics in the Mekong Delta. This is an interesting topic, because the number of floods in this region is increasing. However, several aspects in this paper have to be thoroughly revised because it can be published. This is explained below. Therefore, I recommend a major revision before this paper can be published.

**Response:** We highly appreciate Reviewer#2 for the dedicated reviews and valuable comments on the manuscript. We will address all your comments and substantially revise the manuscript accordingly. Our revisions are described in detail below.

**1. Comment Readability**: The paper is not to-the-point. This paper is about a 1D-hydrodynamic model that has been calibrated and validated for floods in 2011 and 2013. However, the introduction in Chapter 1 is very long and contains many aspects that are not relevant for this study. The same holds for Chapter 2. Also the discussion in Chapter 5 is much too long and should be made to-the-point. Please rewrite Chapters 1 to 6 in a to-the-point way, so that the number of pages will reduce significantly.

**Response:** We will shorten and rewrite the Chapters in to-the-point way in the revised manuscript as suggested by the reviewer.

**2. Comment Description of model set-up**: A crucial aspect is the flooding in cross-sectional direction in this 1D model. A quasi-2D approach is applied for the flood plains. This is explained very briefly and should be explained in detail, because it has a large impact on the model results.

**Response:** We agree with Reviewer#2 to describe more about the methodology of quasi-2D approach in the revised manuscript. We will add and revise text to explain more about the methodology and model technicalities. In addition, an additional figure (Figure S5) will be added to the Supplement 5 to illustrate the quasi-2D modelling method.

Added text: "*The 2D-quasi approach is combined with 1D modelling to simulate the hydraulic dynamics in the floodplains. In the quasi-2D model, the floodplains are described as a network of fictitious river branches and spills with the main rivers. The advantages of the approach include i) transferring characteristics of 2D flow calculations, flow directions into 1D hydrological model; ii) saving computation time due to less requirements on data input; and iii) representing the reliable physical processes in the model (Karl-Erich et al., 2008; Soumendra et al., 2010)*".

Revised text:

*"Floodplains in Cambodia and Vietnam deltas were modelled using two different approaches. The Cambodia floodplain without channels and dikes is simulated by the wide cross-sections using the 1D method. The quasi-2D approach was applied to formulate the hydrodynamic interactions between floodplains, and rivers under dike construction scenarios in the LXQ. Although the Plain of Reeds was not itself the focus of this research, it was also included in the model with the constructed dike in 2011 because there are important hydraulic interactions between the Tien and the Hau Rivers through the Vam Nao River and their tributaries. In these floodplains, there are multitudes of compartments enclosed by dikes and channels. Therefore, each compartment was considered as a flood cell modelled by fictitious river branches with low and wide cross sections extracted from the digital elevation model (DEM) with 90mx90m resolution. These fictitious river branches were linked to real channels by the control structures. Weirs represented dikes and overflows, whereas the dike's height was adjusted by changing the sill-level of control structures. The approach referenced from Dung et al., (2011) is illustrated in Figure S5 in the Supplement."*

*Supplemen 5 and Figure S5:*

[Figure]

*Figure S5: The left figure describes the 1D-quasi2D modelled river network of the VMD and the right figure show a representative typical floodplain compartment. The approach is based on Dung et al., (2011)*

**3. Comment Lack of validation data:** In the abstract and in the conclusions is stated that there is a lack of validation data. However, in Section 3.2 is stated that hourly discharge and water levels are available at 15 locations, of which four locations are even in floodplains. This is a nice validation set. So, there seems to be no lack of validation data.

**Response:** We agree with Reviewer#2 about this point. We did not mean there is a lack of data for simulation and calibration for different locations on the main rivers, but we expected having more data in the floodplain of Cambodia and the downstream part of VMD floodplain. This would have helped us to validate the model performance in these areas as well as to improve accuracy in water balance calculations.

**4. Comment Presentation of model results**: One expects figures with time series of water levels and discharges that contain both numerical results and measurements. However, such

plots are missing. Therefore, a reader does not have any insight whether the time behavior of the floods is simulated accurately. Instead, correlation numbers and maximum high water are presented in the figures. However, this is of secondary importance. The authors are strongly advised to add several time history figures with computed and measured results.

**Response:** We will add time-series of computed and measured water and discharges for several representative stations (Figure 3) in the revised manuscript. Besides, the Q-Q plots and time-series plots of all stations are presented in the Supplement.

Added Figure:

[Figure]

*Figure 3: Time series of simulated and observed flows in 2011 at representative stations*

*In Supplement:*

[Figure]

*Figure S2: Time-series of daily simulated and observed flows in 2011 at all stations used for model calibration*

[Figure]

*Figure S3: Time-series of daily simulated and observed flows in 2013 at all stations used for model calibration*

**5. Comment Description of dykes**: Please clarify the differences between the dyke types (semi-dyke, August dike, high dyke) that are mentioned in this paper.

**Response:** In the revised version of the manuscript, we will clarify differences between the semi-dike and high-dike in the Section 2 (last paragraph). The differences between these two

dike types were also described in the original manuscript, however in the revision we will move the explanation forward in the revised version to help readers understand these types of dike. In addition, we will change "semi-dike" into "low-dike" to help reader understand the term. We will also use the term throughout the manuscript, instead of using "August dike" at some points in the text.

**6. Comment Hinge response**: What is a hinge response? Please clarify.

**Response:** With the Hinge response we want to explain the differences of model results in water level variability between upstream and downstream locations. Water level fluctuations are much higher upstream compared to downstream locations. However, we realize based on the reviews that the use of the term "hinge response" is confusing; therefore, in the revised version of the paper we will not use the term "hinge response" anymore. We will also rename this term in Section 4.

**7. Comment What is new in this paper?** In the discussion (P. 21) is stated that the results are consistent with earlier studies with 1D hydrodynamic models. What is new in this paper? Please add the references of the other studies.

**Response:**

We thank reviewer#2 for his/her suggestions on strengthening the paper's innovation and contribution to current knowledge body. Our study's main innovation and contributions are in assessing changes in floodwater regimes and flow volumes under multiple dike construction scenarios and in adding the water balance component to our modelling analyses. While many previous studies focused on assessing impacts of historic dike developments (Duong et al., 2014; Hoa et al., 2007; Tri et al., 2012), we present one of the first study assessing possible impacts of future dike developments in the Mekong Delta. In addition, our water balance analyses help to better understand the mechanisms of changing flood dynamics due to dike construction. To the best of our knowledge, there are no previous studies reporting results on water balance analyses. Although we agree that our main results are consistent with previous studies, we think that our findings for future dikes development impacts and water balance calculation represent important new contributions. We will revise the result and discussion sections to better highlight these aspects in the manuscript.

**8. Comment Validation for 2000 flood**. Suddenly in the paper the authors start with a validation of the 2000 flood. The results are not very accurate because the geometry of the Mekong Delta was somewhat different in 2000. What is the purpose of this validation? Should this be left out?

**Response:** We agree with Reviewer#2 on differences between the Mekong Delta geometry in different time periods, however we would like to clarify that we do not intend to use the 2000 flood for model validation. We calibrated the model with the 2011 flood and the 2013 flood was used for model validation. We used the flood of 2000 to understand how the model behaves using the historical extreme flood hydrograph in this year. This was done for two main reasons. First, the 2000 flood is one of the major floods happened in the year when very few high-dikes for triple rice production existed in the floodplains. We therefore assumed S1 (the baseline scenario without any high-dike compartment), as similar to as the physical conditions in 2000, to compare the dike impacts with other large-scale dike construction scenarios (S2, S3, and S4). Second, we aim to explore how the model handles the correlation between upstream and

downstream water levels (Tan Chau and Can Tho) in the 2000 and 2011 when the boundary conditions of the 2000 flood were put into the model with existing conditions in 2011. Due to abovementioned reasons, we would like to include the 2000 flood for the comparisons in the manuscript.

**9. Comment Accuracy of model results.** In the discussion (P. 25) is stated that the model results are in line with other studies with 1D model, but that that 2D (and possible) 3D modeling is required for an in-depth understanding of the flood behavior in the Mekong Delta. In other words, do the authors conclude that 1D modeling with a quasi-2D approach for flooding is not suitable for this?

**Response:** It is clear that our model with 1D-quasi2D approach could simulate peak water levels in the dike scenarios of the floodplains, based on good model fitness of calibration and validation results. However, the quasi-2D was just applied to simulate the floodwater interactions mainly for the floodplains of Plain of Reeds and Long Xuyen Quadrangle instead of the context of whole Mekong Delta. We think that the model results could become more accurate if 2D or 3D models will be used for solving complex interactions in the floodplains with the whole river system. Although we acknowledge the potential added values of such 2D and 3D approach, at the moment it is very difficult to pursue such modelling exercise for the whole Mekong Delta due to limited data availability and high computational demands.

**Reference**

Dung, N.V., Merz, B., BĂ¡rdossy, A., Thang, T.D., Apel, H., 2011. Multi-objective automatic calibration of hydrodynamic models utilizing inundation maps and gauge data. Hydrology and Earth System Sciences 15, 1339–1354. doi:10.5194/hess-15-1339-2011

Duong, V.H.T., Trinh Cong, V., Franz, N., Peter, O., Nguyen Trung, N., 2014. Land use based flood hazards analysis for the mekong delta, in: Proceedings of the 19 Th IAHR - APD Congress  2014, Hanoi , Vietnam. Presented at the IAHR, Ha Noi, Vietnam. doi:10.13140/2.1.5153.9842

Hoa, L.T.V., Nguyen, H.N., Wolanski, E., Tran, T.C., Haruyama, S., 2007. The combined impact on the flooding in Vietnam's Mekong River delta of local man-made structures, sea level rise, and dams upstream in the river catchment. Estuarine, Coastal and Shelf Science 71, 110–116. doi:10.1016/j.ecss.2006.08.021

Karl-Erich, L., Huang, S., Baborowski, M., 2008. A quasi-2D flood modeling approach to simulate substance transport in polder systems for environment flood risk assessment. Science of The Total Environment 397, 86–102. doi:http://dx.doi.org/10.1016/j.scitotenv.2008.02.045

Soumendra, N.K., Dhrubajyoti, S., Paul, D.B., 2010. Coupled 1D-Quasi-2D Flood Inundation Model with Unstructured Grids. Journal of Hydraulic Engineering 136, 493–506. doi:10.1061/(ASCE)HY.1943-7900.0000211

Tri, V.P.D., Trung, N.H., Tuu, N.T., 2012. Flow dynamics in the Long Xuyen Quadrangle under the impacts of full-dyke systems and sea level rise. VNU Journal of Science Earth Science 28.

---

## Author Response (AR1)

**Dung Duc Tran et al. Assessing impacts of dike construction on the flood dynamics in the Mekong Delta**

Responses to Reviewer#1 comments

By Dung Duc Tran

dung.ductran@wur.nl or dungtranducvn@yahoo.com

**General comment**: The authors used a hydrology model to evaluate the relationship between dike construction and the hydrography during the flood season in the Vietnamese Mekong Delta. In general, this study is interesting and important, considering the frequency of flood events and high-density dike constructions. However, the impact of artificial construction on flood dynamic obviously has been extensively studied in many previous literature, so I would not say this is a novel study although the numerical modeling method is rarely seen. There are several issues that needed to be addressed before the paper can be accepted in HESS. I recommend a major revision with further review by the editors and referees.

**Response**: We highly appreciate Reviewer#1 for the dedicated reviews and valuable comments on the manuscript. Please find below our details responses and corresponding revisions.

**1. Comment P1, Ln19**: I expect the authors to explain and define some technical terms/words at the first time in the paper, such as the high-dike and semi-dike. In addition, please use the consistent word throughout the paper, for example, correct the August-dike to semi-dike.

**Response:** We agree with Rewiewer#1 to define some key terms (i.e. high-dike, semi-dike) at the beginning of the paper. The high-dikes we considered in the study are the closed polders/compartments used mainly for protecting triple-rice production. We added short text to describe dike types in the Abstract and further elaborated on these terms in the main text.

Added text to the Abstract: "*Accelerated high dike building on the floodplains of the upper delta to allow triple cropping of rice*"

Regarding the semi-dike, we defined it in the Introduction section of the manuscript only because this dike was not considered as a factor causing flood risk downstream. However, we have replaced the "semi-dike" term with "low-dike" term. This term helps readers understand the meaning easily in reference to our use of "high-dike". In addition, the "low-dike" term is used throughout the manuscript to ensure the consistency.

**2. Comment P1, Ln22-23**: This sentence needs to be rephrased:...is assessed through the flood hydrographs modeling under different dike density scenarios in 2011 and 2013.

**Response:** We rephrased the sentence as Reviewer#1's comment in the revised manuscript.

Rephrased text: "*This paper assesses the hydraulic impacts of upstream dike construction on the flood hazard downstream in the Vietnamese Mekong Delta. We combined the existing one-dimensional (1D) Mekong Delta hydrodynamic model with a quasi-two dimensional (2D) approach. First we calibrated and validated the model using flood data from 2011 and 2013. We then applied the model to explore the downstream water dynamics under various scenarios of high dike construction in An Giang Province and the Long Xuyen Quadrangle*".

**3. Comment P2 Ln2**: What is a Quadrangle? I don't think it is a right word for hydrology study. Try to use the watershed name.

**Response:** We agree with the reviewer that using a watershed name is better in a hydrology study. But in our case, "Long Xuyen Quadrangle" term is widely used in literature (Hung et al., 2012, 2014a, 2014b, Manh et al., 2014, 2015; Mekong Delta Plan, 2013), referring to one of the two geographical floodplains in the Vietnamese Mekong Delta (the other floodplain is Plain of Reeds). In addition, Figure 1 shows the geographical extent of the Long Xuyen Quadrangle in the manuscript which helps readers know the location of this floodplain. We also described the Long Xuyen Quadrangle in the Introduction section.

**4. Comment P2, Ln18-20**: I don't suggest to write the future work in the abstract, since it is not part of the authors' work reported in this paper. Also, the authors mentioned that the historical monitoring data are absent, so it is actually difficult or even impossible to do the future assessments.

**Response:** We fully agree with Reviewer#1. These sentences are removed from the Abstract.

**5. Comment P3, Ln2-8**: The information of economic and food production is too detailed within this paper. Try to shorten this part.

**Response:** We agree to shorten this part in the revised manuscript. In particular we have merged two paragraphs about flood benefits and damages into one and remove some unnecessary details.

**6. Comment P3, Ln9-17**: The economic cost and loss are not related to the scientific question in this paper.

**Response:** We agree with the reviewer to remove the content in the revised manuscript.

**7. Comment P5, Ln1-3**: The authors used one point observation to demonstrate a clear correlation between the dike construction and water level. There is no clear evidence showing the cause and effect between the dike construction and water level. In addition, I don't suggest to write one data point in the introduction part.

**Response:** We agree with Reviewer#1 that one point observation could not be used to demonstrate a clear correlation between the dike construction and the increase in water levels. We used this example to highlight a public concern about the flood risk downstream caused by upstream dike construction. It was confused for readers when we used the "suggest there is a clear correlation" phrase in the manuscript. We rephrase this sentence as one of the evidence/example indicating flood risks downstream which would be potentially exacerbated by the large-scale high-dike constructions. This evidence is also presented in a recent study by Triet et al. (2017).

Text are changed as follows: *"At the upstream station of Tan Chau, for example, water levels in 2011 were 0.63 m lower than in 2000 (4.27 m versus 4.90 m). Yet, water levels at the downstream Can Tho station were 0.36 m higher in 2011 than in 2000 (2.15 m versus 1.79 m). This suggests a correlation between the proliferation of dike construction on the floodplains, particularly high dikes, and higher water levels and flood risk downstream"*.

**8. Comment P5, Ln9-24**: I recommend the authors to cite the reference right after each reason of flood risk (Ln10-11), instead of explaining each reference separately in detailed. The other

reasons associated with the flood risks are not strongly related to the scientific questions discussed in this paper. And, are these studies focus on the same study site (Vietnamese Mekong Delta) as well?

**Response:** We thank Reviewer#1 for this useful suggestion. We rewrote this paragraph to better connect references to different reasons of flood risks. We also revised the text to link the information with the objective of the paper. We also added text to inform the study site (VMD) of the referenced papers.

Revised text: *"Several studies have concluded that the flood risk in the VMD delta has increased over time. Numerous reasons have been proposed, such as climate change, sea level rise, hydropower projects, land subsidence and local rainfall. Wassmann et al. (2004) concluded based on a hydraulic model that the higher water levels in the delta were caused by sea level rise in association with climate change. Fujihara et al. (2015) investigated the impacts of upstream runoff, sea level rise and land subsidence on flood levels. They found that flood depths would be significantly increased in 19 tide-dominated areas, and that land subsidence and sea level rise would worsen inundation. Lauri et al. (2012) and Hoang et al. (2016) explored potential impacts of climate change and reservoir management scenarios on the future hydrology of the Mekong River. Numerous authors have considered the effects of climate change and sea level rise on flood propagation, inundated area and sediment transport (Apel et al., 2012; Hung, 2012b; Quang et al., 2012; Manh et al., 2014)."*

**9. Comment P6 Ln22-23**: I suggest the authors highlight the gap of modelling approach within this manuscript. The previous studies of modelling approaches and applications should be addressed in the introduction as well.

**Response:** We highlighted the important gaps of the modelling approach, namely the lack of mechanistic understanding about impacts of upstream high-dike development on downstream flood hazards, and the missing quantifications of the water balance. We added text to emphasize these two gaps in the revised manuscript.

Added text to the revised manuscript: *"Despite the rapid expansion of high dike systems for triple rice cultivation in the upper Mekong Delta, few modelling studies have as yet assessed the implications of such dikes for floodwater regimes. Additionally, most previous studies have focused on changes in peak water levels, based on monitoring data or model results. No study has as yet analyzed the distribution of floodwaters and changes therein. However, water distribution analyses are essential for understanding how floodwaters may spread under different dike construction scenarios."*

**10. Comment P8 Ln4**: What is a.m.s.l?

**Response:** We clarified the term of "a.m.s.l" as "*above mean sea level*".

**11. Comment P9 Ln3-5**: Again, move these explanations forward to help readers understand their meanings.

**Response:** We agree with Reviewer#1 to move these explanations forward to help readers understand the characteristics of low dikes and high dikes. In the revised manuscript, these explanations are moved forward in the third paragraph of Section 2 (**Study area**).

**12. Comment P8 Methodology section**: I think a more detailed introduction of the hydrologic model and software are necessary, including the governing equations and physics used in the model, their applications, pros and cons, etc. The authors can't just cite the references.

**Response:** We added a brief introduction of the hydrologic model and software. In the revised manuscript, we also present detailed introduction in the Appendix.

Added text to the revised manuscript: "*We developed a one-dimensional (1D) hydrodynamic model using the Mike 11 software developed by the Danish Hydraulic Institute (DHI). This is an implicit finite difference model for 1D unsteady flow computation. In addition, it can be applied to a quasi-two dimensional (quasi2D) flow simulation appropriate for detailed modelling of rivers, including special treatment of floodplains, road overtopping, culverts, gate openings and weirs (Doulgeris et al., 2012). The modelling procedure allows use of kinematic, diffusive or fully dynamic, vertically integrated equations for conservation of continuity and momentum (the Saint Venant equations) to solve complex flow and mass transport problems (Patro et al., 2009; Dung et al., 2011; Manh et al., 2014).*

*We developed our model to represent the river network and floodplains of the Mekong Delta. Appendix 1 (A1) presents the equations and computational components.*"

Text added in the Appendix:

*The Saint-Venant equations were formulated as follows* (DHI, 2011)*:*

*Continuity equation:*

$$\frac{\partial Q}{\partial t} + \frac{\partial A}{\partial t} = q$$

*Momentum equation:*

$$\frac{\partial Q}{\partial t} + \frac{\partial \left( \frac{\propto Q^2}{A} \right)}{\partial x} + gA\frac{\partial h}{\partial x} + \frac{gQ|Q|}{C^2 AR} = 0$$

*Where*

*Q-discharge [m³/s]*

*A-flow area [m²]*

*q-the lateral inflow [m²/s]*

*h-stage above datum [m]*

*C-Chezy resistance coefficient [m$^{1/2}$/s]*

*R-hydraulic or resistance radius [m]*

*∝-momentum distribution coefficient*

**13. Comment P11 Ln5-7**: How did you select these Manning coefficients? Any references? And, try to use a clearer word rather than "global" to avoid misunderstanding.

**Response:** We agree to explain our selection of the Manning Coefficients in the revised manuscript and will add references. Beside, "global" term is replaced with "generic" and the

sentence will be clarified to explain our input for a coefficient for the whole rivers in the network.

Explained how to select the Manning Coefficients and references: "*River roughness was represented in the model as Manning coefficients, which we initially estimated based on published values corresponding to particular types of rivers and canals (Chow, 1959; Fabio et al., 2010; Dung et al., 2011)*".

The text will be rewritten as follows: "*First, referring to Chow (1959), we set the Manning coefficients as 0.020 (irrigation channel, straight, on hard-packed smooth sand), 0.025 (earth channel excavated in alluvial silt soil, with deposits of sand on the bottom and grass growth) and 0.033 (natural channel, somewhat irregular side slopes, very little variation in cross section)*"

**14. Comment P14 Ln19**: I'm confused about the Q-Q plots in figure 3. What does each point represent? Are they daily simulated and observed streamflow? If yes, the authors should consider the time-series plot for the streamflow results.

**Response:** Each point in the Q-Q plot in Figure 3 represents the correlation between daily simulated and observed streamflow, based on Correlation Coefficient ($R^2$) and Nash-Sutcliffe Efficiency (NSE).We agree with Reviewer#1 to present some time-series plots in the manuscript instead of the Q-Q plots. Another reason is that Table 1 already shows the NSE and $R^2$. All the Q-Q plots are presented only in the Appendix. We also add all time-series plots to the Appendix. Figure 3 is captioned as "Time-series from simulation and actual flows observed in 2011 at representative stations".

[Figure]

*Figure 3: Time-series from simulation and actual flows observed in 2011 at representative stations*

In Appendix:

[Figure]

*Figure A4: Time-series of daily simulated and observed flows in 2011 at all stations used for model calibration*

[Figure]

*Figure A5: Time-series of daily simulated and observed flows in 2013 at all stations used for model calibration*

**15. Comment P15 Ln12-13**: The x-axis of figure 4 should be the distance instead of the sites. I expect the explanation of underestimated simulated streamflow to be right after the figures.

**Response:** In the submitted manuscript, we used the 4 sites instead the distance shown in the axis because these are important points measured along the Hau River. For example, Chau Doc

and Can Tho are two points we exlore the difference in measured water levels between the floods of 2000 and 2011. In addition, the x-axis of Figure 4 also shows the distance of 4 sites. We added an explanation of underestimated simulated streamflow from Figure 4 (below) in the revised manuscript.

We added an explanation to Figure 4:

[Figure]

Figure 4: Simulated and observed maximum water levels for the 2000, 2011 and 2013 flood years at 4 different stations along the Hau River.

Added text right after the figure: *"Figure 4 shows a good fit between the simulated and observed peak water levels for the floods in 2011 (calibration) and 2013 (validation). In the flood 2000, the fitness is low due to the significant changes in physical topography such as river network and branches and river cross-sections between the model setup of 2011 and the measured data in 2000."*

**16. Comment P15 Ln20**: Should be 2011?

**Response:** We corrected the number to "2011".

**17. Comment P17 Ln5-6, Ln 15**: A general comment: The authors should use more quantitative criteria to demonstrate either the difference or similarity between different scenarios. Statistical measures are highly desired.

**Response:** We thank for the comment of Reviewer#1. We added some statistical calculations to the Appendix to present the difference and similarity between water levels in different dike

scenarios. In addition, we explained whether there are significant differences in the water levels or not in the text of the revised manuscript.

In Appendix:

Table A1: Paired sample test for water level time series along the Hau River in 2011

| Scenario and Difference | N | Mean | Peak | Peak Time | Std. Deviation | 95% Confidence interval of the difference | | t-value | df | p-value |
|---|---|---|---|---|---|---|---|---|---|---|
| **Paired Sample Test for water level (m) time series at Chau Doc** | | | | | | Lower | Upper | | | |
| S1 | 4393 | 2.567 | 3.486 | 12/10/2011 | 0.669 | | | | | |
| S2 | 4393 | 2.908 | 4.152 | 12/10/2011 | 0.882 | | | | | |
| S3 | 4393 | 2.912 | 4.166 | 12/10/2011 | 0.885 | | | | | |
| S4 | 4393 | 2.920 | 4.179 | 12/10/2011 | 0.890 | | | | | |
| Pair S1-S2 | | | | | | -0.374 | -0.308 | -20.415 | 8188 | 0.000 |
| Pair S1-S3 | | | | | | -0.377 | -0.311 | -20.569 | 8175 | 0.000 |
| Pair S1-S4 | | | | | | -0.385 | -0.319 | -20.968 | 8153 | 0.000 |
| **Paired Sample Test for water level (m) time series at Vam Nao** | | | | | | | | | | |
| S1 | 4393 | 1.937 | 2.664 | 13/10/2011 | 0.521 | | | | | |
| S2 | 4393 | 2.030 | 2.943 | 26/10/2011 | 0.583 | | | | | |
| S3 | 4393 | 2.035 | 2.963 | 26/10/2011 | 0.588 | | | | | |
| S4 | 4393 | 2.040 | 2.975 | 26/10/2011 | 0.593 | | | | | |
| Pair S1-S2 | | | | | | -0.116 | -0.070 | -7.914 | 8674 | 0.000 |
| Pair S1-S3 | | | | | | -0.122 | -0.075 | -8.304 | 8656 | 0.000 |
| Pair S1-S4 | | | | | | -0.127 | -0.081 | -8.726 | 8640 | 0.000 |
| **Paired Sample Test for water level (m) time series at Long Xuyen** | | | | | | | | | | |
| S1 | 4393 | 1.654 | 2.431 | 27/10/2011 | 0.499 | | | | | |
| S2 | 4393 | 1.653 | 2.593 | 27/10/2011 | 0.509 | | | | | |
| S3 | 4393 | 1.658 | 2.614 | 26/10/2011 | 0.514 | | | | | |
| S4 | 4393 | 1.664 | 2.625 | 26/10/2011 | 0.519 | | | | | |
| Pair S1-S2 | | | | | | -0.020 | 0.022 | 0.083 | 8780 | 0.934 |
| Pair S1-S3 | | | | | | -0.025 | 0.017 | -0.370 | 8776 | 0.711 |
| Pair S1-S4 | | | | | | -0.031 | 0.012 | -0.862 | 8771 | 0.389 |
| **Paired Sample Test for water level (m) time series at Can Tho** | | | | | | | | | | |
| S1 | 4393 | 0.843 | 2.054 | 27/10/2011 | 0.480 | | | | | |
| S2 | 4393 | 0.829 | 2.098 | 27/10/2011 | 0.499 | | | | | |
| S3 | 4393 | 0.830 | 2.102 | 27/10/2011 | 0.499 | | | | | |
| S4 | 4393 | 0.832 | 2.106 | 27/10/2011 | 0.500 | | | | | |
| Pair S1-S2 | | | | | | -0.006 | 0.035 | 1.368 | 8771 | 0.172 |
| Pair S1-S3 | | | | | | -0.008 | 0.033 | 1.197 | 8770 | 0.231 |
| Pair S1-S4 | | | | | | -0.010 | 0.031 | 1.008 | 8770 | 0.314 |

**18. Comment P17 Ln 20-23**: The authors should provide a more detailed explanation of the "hinge response". A modified hydrograph as forcing should be presented in the paper as well to help readers understand the "hinge response".

**Response:** Based on the responses of both reviewers we have come to the conclusion that the use of the term "hinge response" is confusing. Therefore in the revised version of the paper, we do not use the term "hinge response" anymore. We also rename this section in chapter 4.

**19. Comment P21 Discussion section**: In general, I think the discussion section is too long and verbose. The discussion must be shortened with clearer statements for each analysis. The authors can also try to reconstruct the discussions with results section to help the readers better understand the highlighted study results.

**Response:** We shortened and reconstructed the Discussion section to help readers have better understanding about the significance of the study results.

**20. Comment P26**: Again, a reconstruction of discussion and conclusion sections is needed for this paper. I strongly recommend the authors to use bullets to clearly state the major findings of this study in the conclusion part. Tables and figures should be listed separately. Why did you list the peak water levels in Table 4 instead of plotting in the figures? Probably try to plot the peak water levels under different dike construction as well unless any other reasons.

**Response:** We substantially revised and tightened the structure of the discussion section and create better links to the main results. We also rewrote the Conclusion section by using bullets to state the major findings. Regarding Table 4, we would like to keep it in the revised manuscript because we want to show the detailed differences between peak floodwater levels in different points on canals of the floodplain.

**References**

DHI, 2011. A Modelling System for Rivers and Channels. DHI Software Licence Agreement, Danish Hydraulic Institute.

Doulgeris, C., Georgiou, P., Papadimos, D., Papamichail, D., 2012. Ecosystem approach to water resources management using the MIKE 11 modeling system in the Strymonas River and Lake Kerkini. Journal of Environmental Management 94, 132–143. doi:10.1016/j.jenvman.2011.06.023

Dung, N.V., Merz, B., BĂ¡rdossy, A., Thang, T.D., Apel, H., 2011. Multi-objective automatic calibration of hydrodynamic models utilizing inundation maps and gauge data. Hydrology and Earth System Sciences 15, 1339–1354. doi:10.5194/hess-15-1339-2011

Duong, V.H.T., Trinh Cong, V., Franz, N., Peter, O., Nguyen Trung, N., 2014. Land use based flood hazards analysis for the mekong delta, in: Proceedings of the 19 Th IAHR - APD Congress  2014, Hanoi , Vietnam. Presented at the IAHR, Ha Noi, Vietnam. doi:10.13140/2.1.5153.9842

Fujihara, Y., Hoshikawa, K., Fujii, H., Kotera, A., Nagano, T., Yokoyama, S., 2015. Analysis and attribution of trends in water levels in the Vietnamese Mekong Delta. Hydrological Processes n/a-n/a. doi:10.1002/hyp.10642

Hoang, L.P., Lauri, H., Kummu, M., Koponen, J., van Vliet, M.T.H., Supit, I., Leemans, R., Kabat, P., Ludwig, F., 2016. Mekong River flow and hydrological extremes under climate change. Hydrology and Earth System Sciences 20, 3027–3041. doi:10.5194/hess-20-3027-2016

Hung, N.N., Delgado, J.M., Güntner, A., Merz, B., Bárdossy, A., Apel, H., 2014a. Sedimentation in the floodplains of the Mekong Delta, Vietnam Part II: deposition and erosion. Hydrological Processes 28, 3145–3160. doi:10.1002/hyp.9855

Hung, N.N., Delgado, J.M., Güntner, A., Merz, B., Bárdossy, A., Apel, H., 2014b. Sedimentation in the floodplains of the Mekong Delta, Vietnam. Part I: suspended sediment dynamics. Hydrological Processes 28, 3132–3144. doi:10.1002/hyp.9856

Hung, N.N., Delgado, J.M., Tri, V.K., Hung, L.M., Merz, B., Bárdossy, A., Apel, H., 2012. Floodplain hydrology of the Mekong Delta, Vietnam. Hydrological Processes 26, 674–686. doi:10.1002/hyp.8183

Lauri, H., de Moel, H., Ward, P.J., Räsänen, T.A., Keskinen, M., Kummu, M., 2012. Future changes in Mekong River hydrology: impact of climate change and reservoir operation on discharge. Hydrology and Earth System Sciences 16, 4603–4619. doi:10.5194/hess-16-4603-2012

Manh, N.V., Dung, N.V., Hung, N.N., Kummu, M., Merz, B., Apel, H., 2015. Future sediment dynamics in the Mekong Delta floodplains: Impacts of hydropower development, climate change and sea level rise. Global and Planetary Change 127, 22–33. doi:10.1016/j.gloplacha.2015.01.001

Manh, N.V., Dung, N.V., Hung, N.N., Merz, B., Apel, H., 2014. Large-scale suspended sediment transport and sediment deposition in the Mekong Delta. Hydrology and Earth System Sciences 18, 3033–3053. doi:10.5194/hess-18-3033-2014

Marchand, M., Pham Quang, D., Le, T., 2014. Mekong Delta: Living with water, but for how long? Built Environment 40(2), 230–243.

Mekong Delta Plan, 2013. Kingdom of the Netherlands & The Socialist Republic of Vietnam. Report.

Patro, S., Chatterjee, C., Mohanty, S., Singh, R., Raghuwanshi, N.S., 2009. Flood inundation modeling using MIKE FLOOD and remote sensing data. J Indian Soc Remote Sens 37, 107–118. doi:10.1007/s12524-009-0002-1

Tri, V. P. D., Trung, N.H., Tuu, N.T., 2012. Flow dynamics in the Long Xuyen Quadrangle under the impacts of full-dyke systems and sea level rise. VNU Journal of Science Earth Science 28.

Tri, Van Pham Dang, Trung, N.H., Tuu, N.T., 2012. Flow dynamics in the Long Xuyen Quadrangle under the impacts of full-dyke systems and sea level rise. VNU Journal of Science Earth Science 28.

Triet, N.V.K., Dung, N.V., Fujii, H., Kummu, M., Merz, B., Apel, H., 2017. Has dyke development in the Vietnamese Mekong Delta shifted flood hazard downstream? Hydrology and Earth System Sciences 21, 3991–4010. doi:10.5194/hess-21-3991-2017

Wassmann, R., Hien, N., Hoanh, C., Tuong, T.P., 2004. Sea Level Rise Affecting the Vietnamese Mekong Delta: Water Elevation in the Flood Season and Implications for Rice Production. Climatic Change 66, 89–107. doi:10.1023/B:CLIM.0000043144.69736.b7

**Dung Duc Tran et al. Assessing impacts of dike construction on the flood dynamics in the Mekong Delta**

Responses to Reviewer#2 comments

By Dung Duc Tran

dung.ductran@wur.nl or dungtranducvn@yahoo.com

**General comment**: This paper is about assessing the impacts of dike construction on the flood dynamics in the Mekong Delta. This is an interesting topic, because the number of floods in this region is increasing. However, several aspects in this paper have to be thoroughly revised because it can be published. This is explained below. Therefore, I recommend a major revision before this paper can be published.

**Response:** We highly appreciate Reviewer#2 for the dedicated reviews and valuable comments on the manuscript. We addressed all your comments and substantially revised the manuscript accordingly. Our revisions are described in detail below.

**1. Comment Readability**: The paper is not to-the-point. This paper is about a 1D-hydrodynamic model that has been calibrated and validated for floods in 2011 and 2013. However, the introduction in Chapter 1 is very long and contains many aspects that are not relevant for this study. The same holds for Chapter 2. Also the discussion in Chapter 5 is much too long and should be made to-the-point. Please rewrite Chapters 1 to 6 in a to-the-point way, so that the number of pages will reduce significantly.

**Response:** The manuscript was sent to a professional English editing service to revise all the text. We have shortened and rewrote all the Chapters in to-the-point way in the revised manuscript as suggested by the reviewer.

**2. Comment Description of model set-up**: A crucial aspect is the flooding in cross-sectional direction in this 1D model. A quasi-2D approach is applied for the flood plains. This is explained very briefly and should be explained in detail, because it has a large impact on the model results.

**Response:** We agree with Reviewer#2 to describe more about the methodology of quasi-2D approach in the revised manuscript. We added and revised text to explain more about the methodology and model technicalities. In addition, a figure (Figure A2) is added to the Appendix to illustrate the quasi-2D modelling method.

Added text: "*To simulate the hydraulic dynamics of the floodplains, the quasi2D approach was combined with 1D modelling. In the quasi2D model, the floodplains were described as a network of fictitious river branches and spillovers with the main rivers. This approach had several advantages: (i) transferring some of the benefits of 2D flow calculations and flow directions to the 1D hydrological model; (ii) saving computation time because less input data was needed; and (iii) reliable model representation of physical processes (Karl-Erich et al., 2008; Soumendra et al., 2010)*".

Revised text:

*"We used different approaches to model the floodplains in Cambodia and in Vietnam. The Cambodian floodplains without channels and dikes were simulated by wide cross sections using the 1D method. For the LXQ, we applied the quasi2D approach to formulate the hydrodynamic interactions between the floodplains and rivers under various dike construction scenarios. Although the Plain of Reeds itself was not a focus of this research, we included it in the model with the dikes as constructed in 2011, to better understand the hydraulic interactions between the Tien and Hau rivers via the Vam Nao River and tributaries. The LXQ floodplains are characterized by a dense network of dikes and channels, producing multitudes of compartmentalized fields for agriculture. Each compartment was considered a flood cell and modelled as a fictitious river branch with a low and wide cross section, as extracted from a digital elevation model (DEM, 90 m x 90 m resolution). The control structures linked these fictitious river branches to real channels. Weirs represented dikes and overflows. Dike height was adjusted by changing the sill level of the control structures. This approach, from Dung et al. (2011), is illustrated in Figure A2 in the Appendix."*

*Figure A2 in the Appendix:*

[Figure]

*Figure A2: The left figure describes the 1D-quasi2D modelled river network of the VMD and the right figure show a representative typical floodplain compartment. The approach is from Dung et al. (2011).*

**3. Comment Lack of validation data:** In the abstract and in the conclusions is stated that there is a lack of validation data. However, in Section 3.2 is stated that hourly discharge and water levels are available at 15 locations, of which four locations are even in floodplains. This is a nice validation set. So, there seems to be no lack of validation data.

**Response:** We agree with Reviewer#2 about this point. We did not mean there is a lack of data for simulation and calibration for different locations on the main rivers, but we expected having more data in the floodplain of Cambodia and the downstream part of VMD floodplain. This would have helped us to validate the model performance in these areas as well as to improve accuracy in water balance calculations.

**4. Comment Presentation of model results**: One expects figures with time series of water levels and discharges that contain both numerical results and measurements. However, such

plots are missing. Therefore, a reader does not have any insight whether the time behavior of the floods is simulated accurately. Instead, correlation numbers and maximum high water are presented in the figures. However, this is of secondary importance. The authors are strongly advised to add several time history figures with computed and measured results.

**Response:** We added time-series of computed and measured water and discharges for several representative stations (Figure 3) in the revised manuscript. Besides, the Q-Q plots and time-series plots of all stations are presented in the Appendix.

Added Figure:

[Figure]

*Figure 3: Time series from simulation and actual flows observed in 2011 at representative stations.*

*In Appendix:*

[Figure]

*Figure A4: Time-series of daily simulated and observed flows in 2011 at all stations used for model calibration*

[Figure]

*Figure A5: Time-series of daily simulated and observed flows in 2013 at all stations used for model calibration*

**5. Comment Description of dykes**: Please clarify the differences between the dyke types (semi-dyke, August dike, high dyke) that are mentioned in this paper.

**Response:** In the revised version of the manuscript, we clarified differences between the semi-dike and high-dike in the Section 2 (last paragraph). The differences between these two dike

types were also described in the original manuscript, however in the revision we move the explanation forward (third paragraph in the Section 2) in the revised version to help readers understand these types of dike. In addition, we change "semi-dike" into "low-dike" to help reader understand the term. We also use the term throughout the manuscript, instead of using "August dike" at some points in the text.

**6. Comment Hinge response**: What is a hinge response? Please clarify.

**Response:** With the Hinge response we want to explain the differences of model results in water level variability between upstream and downstream locations. Water level fluctuations are much higher upstream compared to downstream locations. However, we realize based on the reviews that the use of the term "hinge response" is confusing; therefore, in the revised version of the paper we do not use the term "hinge response" anymore. We also renamed this term in Section 4.

**7. Comment What is new in this paper?** In the discussion (P. 21) is stated that the results are consistent with earlier studies with 1D hydrodynamic models. What is new in this paper? Please add the references of the other studies.

**Response:**

We thank reviewer#2 for his/her suggestions on strengthening the paper's innovation and contribution to current knowledge body. Our study's main innovation and contributions are in assessing changes in floodwater regimes and flow volumes under multiple dike construction scenarios and in adding the water balance component to our modelling analyses. While many previous studies focused on assessing impacts of historic dike developments (Duong et al., 2014; Hoa et al., 2007; Tri et al., 2012), we present one of the first study assessing possible impacts of future dike developments in the Mekong Delta. In addition, our water balance analyses help to better understand the mechanisms of changing flood dynamics due to dike construction. To the best of our knowledge, there are no previous studies reporting results on water balance analyses. Although we agree that our main results are consistent with previous studies, we think that our findings for future dikes development impacts and water balance calculation represent important new contributions. We have rewritten and reconstructed the result and discussion sections to better highlight these aspects in the revised manuscript.

**8. Comment Validation for 2000 flood**. Suddenly in the paper the authors start with a validation of the 2000 flood. The results are not very accurate because the geometry of the Mekong Delta was somewhat different in 2000. What is the purpose of this validation? Should this be left out?

**Response:** We agree with Reviewer#2 on differences between the Mekong Delta geometry in different time periods, however we would like to clarify that we do not intend to use the 2000 flood for model validation. We calibrated the model with the 2011 flood and the 2013 flood was used for model validation. We used the flood of 2000 to understand how the model behaves using the historical extreme flood hydrograph in this year. This was done for two main reasons. First, the 2000 flood is one of the major floods happened in the year when very few high-dikes for triple rice production existed in the floodplains. We therefore assumed S1 (the baseline scenario without any high-dike compartment), as similar to as the physical conditions in 2000, to compare the dike impacts with other large-scale dike construction scenarios (S2, S3, and S4). Second, we aim to explore how the model handles the correlation between upstream and

downstream water levels (Tan Chau and Can Tho) in the 2000 and 2011 when the boundary conditions of the 2000 flood were put into the model with existing conditions in 2011. Due to abovementioned reasons, we would like to include the 2000 flood for the comparisons in the manuscript.

**9. Comment Accuracy of model results.** In the discussion (P. 25) is stated that the model results are in line with other studies with 1D model, but that that 2D (and possible) 3D modeling is required for an in-depth understanding of the flood behavior in the Mekong Delta. In other words, do the authors conclude that 1D modeling with a quasi-2D approach for flooding is not suitable for this?

**Response:** It is clear that our model with 1D-quasi2D approach could simulate peak water levels in the dike scenarios of the floodplains, based on good model fitness of calibration and validation results. However, the quasi-2D was just applied to simulate the floodwater interactions mainly for the floodplains of Plain of Reeds and Long Xuyen Quadrangle instead of the context of whole Mekong Delta. We think that the model results could become more accurate if 2D or 3D models will be used for solving complex interactions in the floodplains with the whole river system. Although we acknowledge the potential added values of such 2D and 3D approach, at the moment it is very difficult to pursue such modelling exercise for the whole Mekong Delta due to limited data availability and high computational demands.

**Reference**
Dung, N.V., Merz, B., BĂ¡rdossy, A., Thang, T.D., Apel, H., 2011. Multi-objective automatic calibration of hydrodynamic models utilizing inundation maps and gauge data. Hydrology and Earth System Sciences 15, 1339–1354. doi:10.5194/hess-15-1339-2011

Duong, V.H.T., Trinh Cong, V., Franz, N., Peter, O., Nguyen Trung, N., 2014. Land use based flood hazards analysis for the mekong delta, in: Proceedings of the 19 Th IAHR - APD Congress 2014, Hanoi , Vietnam. Presented at the IAHR, Ha Noi, Vietnam. doi:10.13140/2.1.5153.9842

Hoa, L.T.V., Nguyen, H.N., Wolanski, E., Tran, T.C., Haruyama, S., 2007. The combined impact on the flooding in Vietnam's Mekong River delta of local man-made structures, sea level rise, and dams upstream in the river catchment. Estuarine, Coastal and Shelf Science 71, 110–116. doi:10.1016/j.ecss.2006.08.021

Karl-Erich, L., Huang, S., Baborowski, M., 2008. A quasi-2D flood modeling approach to simulate substance transport in polder systems for environment flood risk assessment. Science of The Total Environment 397, 86–102. doi:http://dx.doi.org/10.1016/j.scitotenv.2008.02.045

Soumendra, N.K., Dhrubajyoti, S., Paul, D.B., 2010. Coupled 1D-Quasi-2D Flood Inundation Model with Unstructured Grids. Journal of Hydraulic Engineering 136, 493–506. doi:10.1061/(ASCE)HY.1943-7900.0000211

Tri, V.P.D., Trung, N.H., Tuu, N.T., 2012. Flow dynamics in the Long Xuyen Quadrangle under the impacts of full-dyke systems and sea level rise. VNU Journal of Science Earth Science 28.

---

## Referee Report (RR1)

HESS-2017-141 – Review of "Assessing impacts of dike construction on the flood dynamics of the Mekong"

In this paper, the authors assess the hydrologic and hydraulic impacts of levee (dike) expansion on flooding within the Vietnamese portion of the Mekong Delta. The authors employ an existing Mike 11 model which uses a quasi-two-dimensional approach to assess changes in water-surface elevations (WSELs) and the water balance within the delta for four levee configuration scenarios. The result of the modeling showed levees increased WSELs upstream of the levee construction and their water balance analyses suggests a substantial amount of river discharge has been or will be diverted away from the wetlands of the Long Xuyen Quadrangle.

Overall, the paper is reasonably well written, and the manuscript should be considered for publication after a few minor issues are addressed. Please see my comments and suggestions below.

**General Comments:**

1. The most interesting finding in this study are the changes in the delta's water balance. The finding that levees impact WSELs upstream of the levee constriction is less interesting because it is: (1) predicted by hydraulic theory (e.g, Yen, 1995 and Akan, 2006); (2) has been documented in empirical studies (e.g., Hiene and Pinter, 2011); and (3) several modeling studies around the world. Focusing the paper on changes in the water balance and discussing the implications of these changes for river management would make this manuscript stronger.
2. Somewhere in the discussion section the authors should provide a relatively brief caveat about the limitations of their hydraulic model related to changes in the distribution of flow in the delta, the potential associated geometric channel changes, and the possible impact (i.e., uncertainty) on their model predictions of WSELs and water volume estimates.
3. Caveats about the limitations of their model should not be included in the abstract. Dialogue about the limitation of their models are best suited to the discussion section of the paper.

**Specific Comments:**

1. Abstract - Line 8 The term "river levels" is confusing in this context; Please specify you are talking about river discharges and not WSELs here.
2. Page 4 - Lines 8-10 The assertion that the change in WSEL for two floods is a "correlation" attributed to levee construction is not appropriate given the natural variability in the stage discharge relationship of a sand-bed and tidally-influenced river. I suggest using the word comparison verse correlation. Statistically speaking, you cannot make a correlation between to observations.
3. Page 5 – Lines 12-20 Please clarify what the authors mean by "floodwater regimes".
4. Page 5 – Lines 21-24 The authors should include a sentence or two here why distribution analysis is essential. In addition, the authors should clarify what they mean by "distribution" analyses. Are distribution analyses the same thing as water balance analyses?
5. Page 6 – Line 10 and 11. The sentence starting with "This paper presents" is superfluous and should be removed.
6. Page 8 Within the modeling setup description more detail is needed about the quasi-two-dimensional cells. Specifically, how were the cell extents defined and what elevation data were used to define the cell volume?

7. Page 12 Please quantify what the authors mean by "high dikes".
8. Page 19 – Line 1 Please use consistent terminology to describe the modeling scenarios (i.e., S1, S2, and S3)
9. Page 20 – Line 4 A space is needed between "(over) compensated".
10. Page 21 – Lines 13-15 As worded, these discussion points seem in conflict with bullet point three in the conclusion section. Please specify *where* future levee expansion or heightening will have little to no impact on WSELs (i.e., downstream of the levee constriction). In addition, raising the elevation for dikes would likely increase the WSELs for floods which would have a large enough magnitude to overtop the current levees. Unless, the levee constricted flow resulted in substantial channel-bed scour and consequently increase the channel's flood water carrying capacity resulting in no change or possibility a reduction in the WSEL.
11. Page 21 – Line 19 Would cubic kilometers be a more appropriate unit for the estimated flood volumes here?
12. Page 22 – Line 15 A comma is needed - "In part, this"
13. Page 23 – Line 2 Is the estimated increase of the WSEL at Can Tho based off the modeling performed in this study or the work of Hoa et al., 2007. Please specify the source for this estimate.
14. Page 23 - Line 7 and 8 The relative stability of discharge in the lower reached of the Hau River may be the *likely* explanation for stability of WSELs at Can Tho. However, it is not the only explanation unless the authors have bathymetric data showing there was no substantial changes in channel geometry (i.e., scour) between the temporal points of comparison.
15. Page 24 – Line 22   I believe the authors mean *hydraulic,* not hydrologic impact.
16. Page 23 – Lines 18 and 19 I recommend using cubic kilometers instead of cubic meters.
17. Page 25 – Lines 1 and 2 Again, this finding is consistent with hydraulic theory and not unexpected.
18. Page 25 – Line 3 and 4 It is not clear to the reviewer how continued levee expansion will increase flood risk across the entire LQX. Does the estimated additional 100 cm of WSEL include just levee impacts or the cumulative effects of levees, sedimentation, and sea level rise? Based on the discussion in lines 8 to 16 on page 25, I believe the authors mean continued levee construction will "likely exacerbate flood risk". As this sentence is currently worded, it seems the authors are attributing the entire 100 cm of the anticipated increase in WSELs within the LQX to future levee expansion.
19. Page 25 – lines 6 and 7 It is not clear what the authors are inferring here. Are the authors suggesting levees are increasing river discharges or are the differences in discharge attributable to model uncertainty? While it is possible for levees to increase river discharges by reducing static and transient water storage through the confinement of flood flows to a levee-defined floodway, such an assertion should be laid out in the discussion section stating how this is theoretically possible.
20. Page 25 – Line 14 – Do the authors mean hydraulic modeling perspective? Hydrological models are not commonly use to assess levee impacts on WSELs.
21. Figure 5 – Abbreviations such as LQX should be defined in the figure notes or caption.
22. Figures 6 and 7 The y-axis units are confusing and not consistent with Figure 8. I recommend putting the units in meters or centimeters in all three figures.

23. Figure 9 – Reporting the water balance scenario differences in percent change would make these changes appear more substantial and not "radical".

---

## Author Response (AR2)

**Dung Duc Tran et al. Assessing impacts of dike construction on the flood dynamics in the Mekong Delta**

Responses to Reviewer#1 comments

By Dung Duc Tran

dung.ductran@wur.nl or dungtranducvn@yahoo.com

**General comment**: I think the authors have addressed most comments and concerns I made in the last revision. The quality of this paper is significantly improved. I have a few additional minor comments below. And, a copy-editing is still required for some English issues.

We highly appreciate Reviewer#1 for the dedicated reviews and valuable comments on the manuscript. Please find below our detail responses. All the revision is present in red color in the main text of the revised manuscript.

**Specific Comments**

**Comment 1.** I don't recommend to put the equations and computational components in the Appendix. They should be introduced in the methodology section. And, I really want to see further and detailed information in the methodology, eg., how does the model determine the exchanges between river and floodplain?

**Response:** To address this issue we added an additional paragraph to the methodology to explain in more details the exchange between rivers and floodplains. We would like to keep the detailed equations in the annex to limit the length of the paper. Also, the equations have been described in detail in previous papers by Hoa et al, (2008), Soumendra et al., (2010), and Tri et al., (2012).

**Comment 2.** And also, I think the time-series plots should be move to the result section, too. I don't understand why authors decide to leave them in the appendix.

**Response:** We moved the time-series plots to the result section as recommended by the reviewer.

**Comment 3.** It seems the numbers of hours above threshold in Table 2 are observation results but not the simulation results. How is the model result?

**Response:** Table 2 presents the tidal water levels in numbers of hours above the threshold as the backwater effects at the two coastal stations in the two flood years of 2000 and 2011. We present the observed tidal water levels to better understand phenomenon of the backwater effect; therefore, we did not analyze the model results.

**Comment 4.** Using 2011 river and infrastructure network for 2000 simulation seems worrisome for me. Is it possible to make any estimation for 2000 dike developments and river network?

**Response:** We understand the concern of the reviewer about the use of 2011 river and infrastructure network for 2000 simulation, but we did not use it to answer the main objective of this paper. We used it to test the performance of the available hydraulic model in simulating the peak water levels of the 2000 flood for discussion. The main objective of our study is to

present the impact of dike construction scenarios based on the calibration and verification from the 2011 and 2013 floods.

**Comment 5.** I doubt if the author can claim "tidal backwater effect is minimal" from the 0.08 m increase, considering the 0.28 m change in river and infrastructure network. 0.08 is not minimal to 0.28.

**Response:** We changed the sentence "tidal backwater effect seems to be limited" instead of "tidal backwater effect is minimal" which makes confusing for readers.

**Comment 6.** I feel the discussion section is still long and verbose. Try to address the benefits of this model in simulating river discharge in dike-developed scenarios. Eg. how does the model improve? Does it compare with other models?

**Response:** Compared to the first version of manuscript, we shortened this part significantly as the comments of the reviewer, but kept essential information to present our main findings. As the comment of the reviewer, we added 2 sentences in the first paragraph of the discussion section to present the benefit of the model. We compared the findings on the impacts of dike construction with previous studies. Please find these in the revised manuscript.

**Comment 7.** In the end of the response to reviewer #2, the authors mentioned the computational demands are one of the major limits for 2D and 3D approach. Could you provide an estimation of computational cost for this simulation?

**Response:** We cannot estimate the exact computational cost for the simulation of 2D and 3D approach applied for the Mekong Delta but based on simulations for part of the delta we know it is very high. For a large and complex river network as the Mekong Delta, it takes many hours for one simulation with our 1D hydraulic model. In addition, the model is unstable during the simulations if there are errors in the input data. We developed a 2D model network for one of the two floodplains of the Mekong Delta (Long Xuyen Quadrangle), and it took several days for one run using a strong computer.

**References**

Hoa, L.T.V., Shigeko, H., Nhan, N.H., Cong, T.T., 2008. Infrastructure effects on floods in the Mekong River Delta in Vietnam. Hydrological Processes 22, 1359–1372. https://doi.org/10.1002/hyp.6945

Soumendra, N.K., Dhrubajyoti, S., Paul, D.B., 2010. Coupled 1D-Quasi-2D Flood Inundation Model with Unstructured Grids. Journal of Hydraulic Engineering 136, 493–506. https://doi.org/10.1061/(ASCE)HY.1943-7900.0000211

Tri, V.P.D., Trung, N.H., Tuu, N.T., 2012. Flow dynamics in the Long Xuyen Quadrangle under the impacts of full-dyke systems and sea level rise. VNU J. Sci. Earth Science 28.

**Dung Duc Tran et al. Assessing impacts of dike construction on the flood dynamics in the Mekong Delta**

Responses to Reviewer#3 comments

By Dung Duc Tran

dung.ductran@wur.nl or dungtranducvn@yahoo.com

HESS-2017-141 – Review of "Assessing impacts of dike construction on the flood dynamics of the Mekong"

In this paper, the authors assess the hydrologic and hydraulic impacts of levee (dike) expansion on flooding within the Vietnamese portion of the Mekong Delta. The authors employ an existing Mike 11 model which uses a quasi-two-dimensional approach to assess changes in water-surface elevations (WSELs) and the water balance within the delta for four levee configuration scenarios. The result of the modeling showed levees increased WSELs upstream of the levee construction and their water balance analyses suggests a substantial amount of river discharge has been or will be diverted away from the wetlands of the Long Xuyen Quadrangle.

Overall, the paper is reasonably well written, and the manuscript should be considered for publication after a few minor issues are addressed. Please see my comments and suggestions below.

**General Comments**
1. The most interesting finding in this study are the changes in the delta's water balance. The finding that levees impact WSELs upstream of the levee constriction is less interesting because it is: (1) predicted by hydraulic theory (e.g, Yen, 1995 and Akan, 2006); (2) has been documented in empirical studies (e.g., Hiene and Pinter, 2011); and (3) several modeling studies around the world. Focusing the paper on changes in the water balance and discussing the implications of these changes for river management would make this manuscript stronger.

2. Somewhere in the discussion section the authors should provide a relatively brief caveat about the limitations of their hydraulic model related to changes in the distribution of flow in the delta, the potential associated geometric channel changes, and the possible impact (i.e., uncertainty) on their model predictions of WSELs and water volume estimates.

3. Caveats about the limitations of their model should not be included in the abstract. Dialogue about the limitation of their models are best suited to the discussion section of the paper.

**Response:** We highly appreciate Reviewer#3 for the dedicated reviews and valuable comments on the manuscript. To address points 2 and 3 we re-moved the limitation from the abstract and added a part to the discussion on the model caveat. The general remark one is well noted but would require a major revision at this point. Replies to the specific comments are shown below and in red color in the main text of the revised manuscript.

**Specific Comments**

**Comment 1. Abstract - Line 8** The term "river levels" is confusing in this context; Please specify you are talking about river discharges and not WSELs here.
**Response:** We removed the term "higher" in the sentence to avoid the confusion for readers in the context but still use the terms "river levels".

**Comment 2. Page 4 - Lines 8-10** The assertion that the change in WSEL for two floods is a "correlation" attributed to levee construction is not appropriate given the natural variability in the stage discharge relationship of a sand-bed and tidally-influenced river. I suggest using the word comparison verse correlation. Statistically speaking, you cannot make a correlation between to observations.

**Response:** We replaced the term "correlation" by "relationship".

**Comment 3. Page 5 – Lines 12-20** Please clarify what the authors mean by "floodwater regimes".

**Response:** We clarified the term "floodwater regimes" by adding a footnote. Please find our addition in the revised manuscript.

**Comment 4. Page 5 – Lines 21-24** The authors should include a sentence or two here why distribution analysis is essential. In addition, the authors should clarify what they mean by "distribution" analyses. Are distribution analyses the same thing as water balance analyses?

**Response:** We added a sentence in the revised manuscript to indicate why the distribution analysis is essential (Page 5, lines 18-19). This added sentence could also clarify what the distribution analyses are.

**Comment 5. Page 6 – Line 10 and 11**. The sentence starting with "This paper presents" is superfluous and should be removed.

**Response:** We rewrote the sentence.

**Comment 6. Page 8** Within the modeling setup description more detail is needed about the quasi-two-dimensional cells. Specifically, how were the cell extents defined and what elevation data were used to define the cell volume?

**Response:** We added and rephrased one paragraph in the revised manuscript (Page 12, lines 7-18) to describe how the cell extents were defined and elevation data were used to define the cell volume in the methodology.

**Comment 7. Page 12** Please quantify what the authors mean by "high dikes".

**Response:** We made it by inserting a footnote as explanation. Please find in the revised manuscript.

**Comment 8. Page 19 – Line 1** Please use consistent terminology to describe the modeling scenarios (i.e., S1, S2, and S3)

**Response:** Thank the reviewer for your comment. We checked the consistency in using these terminologies in the revised manuscript. Therefore, we replaced 2, 3, and 4 to be S2, S3, and S4 (line 1 and line 9, page 19 of the manuscript).

**Comment 9. Page 20 – Line 4** A space is needed between "(over) compensated".

**Response:** A space is added.

**Comment 10. Page 21 – Lines 13-15** As worded, these discussion points seem in conflict with bullet point three in the conclusion section. Please specify where future levee expansion or heightening will have little to no impact on WSELs (i.e., downstream of the levee constriction).

In addition, raising the elevation for dikes would likely increase the WSELs for floods which would have a large enough magnitude to overtop the current levees. Unless, the levee constricted flow resulted in substantial channel-bed scour and consequently increase the channel's flood water carrying capacity resulting in no change or possibility a reduction in the WSEL.

**Response:** We added a sentence in the same paragraph of discussion section (lines 16 and 17, page 21 in red color in the revised manuscript) to clarify these points to avoid the confusion to readers. In the discussion, we stated the small impact on river levels from the expansion of high dikes in case the impact is compared to the existing dike condition of 2011 (S2). In the conclusion, however, the impact is stated as the comparison between the dike expansion from 2011 onwards (S2, S3, and S4) compared with the situation of 2000 (S1-without high-dikes).

The new sentence now reads: "Compared to the dike construction in 2011 (scenario S2), additional expansion of dikes is thus expected to have only small additional impact on river levels".

**Comment 11. Page 21 – Line 19** Would cubic kilometers be a more appropriate unit for the estimated flood volumes here?

**Response:** We appreciate Reviewer#3 for the comment, but we still want to keep it in cubic meter and meter for the whole paper to be consistent

**Comment 12. Page 22 – Line 15** A comma is needed - "In part, this"

**Response:** We add a comma to the revised manuscript.

**Comment 13. Page 23 – Line 2** Is the estimated increase of the WSEL at Can Tho based off the modeling performed in this study or the work of Hoa et al., 2007. Please specify the source for this estimate.

**Response:** We added the reference Hoa et al., 2007 to the last part of the sentence.

**Comment 14. Page 23 - Line 7 and 8** The relative stability of discharge in the lower reached of the Hau River may be the likely explanation for stability of WSELs at Can Tho. However, it is not the only explanation unless the authors have bathymetric data showing there was no substantial changes in channel geometry (i.e., scour) between the temporal points of comparison.

**Response:** We agree with the reviewer's comment. We have no detailed bathymetric data but we do have cross-sections at this point to ensure that there are no substantial changes in channel geometry. In our study, the stable surface water levels at Can Tho shown in both the observed and simulated data, could be affected by the two issues i) water delivering to branches along the main rivers before this point to the West Sea, and ii) two opposite water forces at Can Tho from upstream flow and tide from the sea.

**Comment 15. Page 24 – Line 22** I believe the authors mean hydraulic, not hydrologic impact.

**Response:** The reviewer is correct we changed hydrologic to hydraulic

**Comment 16. Page 23 – Lines 18 and 19** I recommend using cubic kilometers instead of cubic meters.

**Response:** We would like to continue to use cubic meters to ensure a consistent unit throughout the paper.

**Comment 17. Page 25 – Lines 1 and 2** Again, this finding is consistent with hydraulic theory and not unexpected.

**Response:** We agree with the reviewer that this finding is consistent with hydraulic theory but what we try to stress in these sentences the main difference between upstream and downstream rivers in their response to the dike construction scenarios.

**Comment 18. Page 25 – Line 3 and 4** It is not clear to the reviewer how continued levee expansion will increase flood risk across the entire LQX. Does the estimated additional 100 cm of WSEL include just levee impacts or the cumulative effects of levees, sedimentation, and sea level rise? Based on the discussion in lines 8 to 16 on page 25, I believe the authors mean continued levee construction will "likely exacerbate flood risk". As this sentence is currently worded, it seems the authors are attributing the entire 100 cm of the anticipated increase in WSELs within the LQX to future levee expansion.

**Response:** This is additional 100 cm of WSEL due to the dike scenario impact which the authors want to state, but we made a confusion to readers. We added a phrase "over the period from 2000" to conclude from the findings of the study. Please find the added text in red color in the revised manuscript.

**Comment 19. Page 25 – lines 6 and 7** It is not clear what the authors are inferring here. Are the authors suggesting levees are increasing river discharges or are the differences in discharge attributable to model uncertainty? While it is possible for levees to increase river discharges by reducing static and transient water storage through the confinement of flood flows to a levee-defined floodway, such an assertion should be laid out in the discussion section stating how this is theoretically possible.

**Response:** The expansion of dike construction in the Long Xuyen Quadrangle floodplain increases the flood river discharges. Due to the dike construction, the floodplain has reduced its retention volume, which is then stored on the rivers, to increase the discharges. Our conclusion points in lines 6 and 7 of the page 19 results from our model findings which discuss about the excess of the retention volume lost due to dike construction (paragraphs 2 and 3 of the discussion section).

**Comment 20. Page 25 – Line 14 –** Do the authors mean hydraulic modeling perspective? Hydrological models are not commonly use to assess levee impacts on WSELs.

**Response:** We mean hydraulic modeling perspective. Please find our change in the revised manuscript.

**Comment 21.** Figure 5 – Abbreviations such as LQX should be defined in the figure notes or caption.

**Response:** We defined LXQ in Figure 5.

**Comment 22.** Figures 6 and 7 The y-axis units are confusing and not consistent with Figure 8. I recommend putting the units in meters or centimeters in all three figures.

**Response:** We changed the y-axis unit in Figure 8 to be centimeter. The units are now consistent in centimeters in all three figures.

**Comment 23.** Figure 9 – Reporting the water balance scenario differences in percent change would make these changes appear more substantial and not "radical".

**Response:** We highly appreciate Reviewer#3 for the valuable comments. However, we prefer to present the increased or decreased numbers in floodwater volume instead of percentage because readers could understand specific numbers as quantitative dynamics of the floodwater volumes under the dike scenarios.

---

## Author Response (AR3)

Dear Editor (Dr. Dominic Mazvimavi),

We would like to thank for accepting our manuscript for publication with some technical corrections needed. Based on the editor's comments, we revise in the manuscript in numbered orders as follows.

**Editor Decision: Publish subject to technical corrections**

(10 Feb 2018) by Dr. Dominic Mazvimavi

Comments to the Author:

There are minor editorial corrections required. The manuscript should not have Appendices. All the material should be presented where appropriate (1). Equations given as Appendix 1 should be incorporated in the main text (2). Similarly diagrams and tables referred to as being in Appendices should be in the main text (3).

The inclusion of some diagrams with numbering that refers to Appendices and other diagrams without such numbering will cause confusion to readers (4).

river level should be river water level throughout. There is a different between a river level and river water level (6).

Avoid using of colon unless you really understand how to use this correctly (7).
* * *
**Authors' response**

(1) & (3). We moved Appendices to the main text, just after the Conclusion section of the manuscript. We renamed Appendix 1, 2, 3, and 4 into Appendix A, B, and C.

(2). We incorporated the equations given as Appendix 1 (from the old version) into the main text (in the correction version, paragraph 1 of heading 3.1 of Methodology section).

(4). We do not refer specific figures/tables of the Appendix as did in the old version of the manuscript. In this correction version, we refer Appendix A, B, and C to avoid confusion for readers.

(5). We checked throughout the manuscript to rephrase "river level" to be "river water level".

(6). The manuscript has been edited by a native English editor from a professional editing company. The using of colon in the old version of the manuscript is revised.

Best regards,
Dung Duc Tran
Corresponding author.